# The Grb2 splice variant, Grb3-3, is a negative regulator of RAS activation

Caroline Seiler[1,5], Amy K. Stainthorp[1,5], Sophie Ketchen [1,5], Christopher M. Jones [1,2,3], Kate Marks[4], Philip Quirke [4] & John E. Ladbury [1✉]

Activation of RAS is crucial in driving cellular outcomes including proliferation, differentiation, migration and apoptosis via the MAPK pathway. This is initiated on recruitment of Grb2, as part of a Grb2-Sos complex, to an up-regulated receptor tyrosine kinase (RTK), enabling subsequent interaction of Sos with the plasma membrane-localised RAS. Aberrant regulation at this convergence point for RTKs in MAPK signalling is a key driver of multiple cancers. Splicing of the *GRB2* gene produces a deletion variant, Grb3-3, that is incapable of binding to RTKs. We show that, despite maintaining the ability to bind to Sos, the Grb3-3-Sos complex remains in the cytoplasm, unable to engage with RAS. Competition between Grb2 and Grb3-3 for binding to C-terminal proline-rich sequences on Sos modulates MAPK signalling. Additionally, we demonstrate that splicing is regulated by heterogenous nuclear riboproteins C1/C2, and that normal and malignant colon tissue show differential Grb3-3 expression.

[1] School of Molecular and Cellular Biology, University of Leeds, Leeds, UK. [2] Radiotherapy Research Group, Faculty of Medicine & Health, University of Leeds, Leeds, UK. [3] Leeds Cancer Centre, The Leeds Teaching Hospitals NHS Trust, Leeds, UK. [4] University of Leeds School of Medicine, Leeds Institute of Medical Research, Pathology and Data Analytics, University of Leeds, Leeds, UK. [5]These authors contributed equally: Caroline Seiler, Amy K. Stainthorp, Sophie Ketchen. ✉email: j.e.ladbury@leeds.ac.uk

Mitogen-activated protein kinase (MAPK, aka. ERK) signal transduction pathways play key roles in regulating essential cellular processes such as proliferation, differentiation, migration and apoptosis. The RAS/RAF/MEK/extracellular signal-related kinase (ERK) cascade is the best characterised of the mammalian MAPK pathways and its constitutive activation is implicated in the development and progression of a majority of cancers[1].

Signalling via this pathway (hereafter termed the MAPK pathway) is upregulated by activation of receptor tyrosine kinases (RTKs)[2,3]. MAPK signalling occurs, in the first instance, when the guanine nucleotide exchange factor, Son of Sevenless (Sos), is recruited to a complex with an RTK at the plasma membrane. This complex is mediated by the adaptor protein, growth factor receptor bound protein 2 (Grb2). Grb2 consists of two Src homology 3 (SH3) domains that sequentially sandwich a SH2 domain. The SH3 domain(s) bind proline-rich sequences (minimally requiring the sequence PXXP, where X is any amino acid) on Sos, whilst the SH2 domain is recruited to phosphorylated tyrosine (pY) residues on RTKs[4,5]. The predominant interaction between Grb2 and Sos appears to be through the N-terminal SH3 domain (NSH3) of the adaptor protein[6]. Once localised at the plasma membrane, Sos binds to membrane-anchored RAS and catalyses the exchange of RAS-GDP for RAS-GTP. The GTP-bound state of RAS then interacts with the RAS-binding domains (RBDs) of RAF kinases, which are consequentially up-regulated to promote downstream signalling.

In most cancers, constitutive activation of the MAPK pathway results from hyperactivity of RTKs or RAS[7]. In the case of RTKs, this can be achieved through receptor overexpression, mutation or sustained autocrine/paracrine production of RTK-activating ligands. The consequential prolonged availability of pY-containing sites enhances recruitment of the Grb2-Sos complex and up-regulation of MAPK signalling. RAS hyperactivity rarely results from overexpression and is predominantly the result of activating mutations. Despite having no intrinsic kinase activity, there is some evidence that overexpression of Grb2 can also mediate MAPK activation in a number of malignancies, including colon cancer[8–10]. RAS activation is a critical determinant of MAPK signalling and, together with Grb2-Sos, constitutes a point of convergence for multiple RTKs[11]. There are no effective inhibitors of the interactions at this critical initiation point of MAPK signalling in routine clinical use[7]. Instead, in RAS-wild type cancers individual or even multiple RTKs are targeted based on their overexpression or activation[12].

A natural human isoform of Grb2, Grb3-3, arises through alternate splicing of exon four of the *GRB2* gene[13]. The resulting protein is characterised by the deletion of 40 residues of the Grb2 SH2 domain, though both functional SH3 domains are retained. Corruption of the SH2 domain abrogates pY binding, and hence recruitment to RTKs. However, the SH3 domains remain capable of binding proline-rich sequences. Little is known about the cellular role of Grb3-3, though it is implicated in apoptosis and is suggested to act to abrogate RTK-stimulated Grb2-mediated RAS activation[13–15].

Although, signalling cascades driven by RAS are vital to normal cellular function, they are also responsible for the development and progression of multiple cancers. Consequently, the understanding of mechanisms governing RAS activation is of considerable interest, since it may identify promising, novel routes to therapeutic intervention. Here, we report a previously unrecognised mechanism for negative regulation of RAS activation by the Grb2 splice variant Grb3-3. This plays a tumour-suppressive role by competing with Grb2 to bind Sos. Where Grb3-3 prevails, RTK-induced activation of wild-type MAPK signalling and downstream cellular proliferation is abrogated;

including in colon cancer. Following a screening investigation we were able to demonstrate that alternate splicing of *GRB2* is regulated by heterogenous nuclear ribonucleoproteins C1/C2 (hnRNPC) which suggests a potential therapeutically relevant route to exert control over Grb2/Grb3-3 expression.

## Results

**Grb3-3 inhibits growth factor-induced RAS-ERK signalling and cell proliferation.** The central role that Grb2 plays in convergence of RTKs to mediate MAPK signalling suggests that, since it has a corrupted SH2 domain, over-expression of Grb3-3 will block the recruitment of Sos to RTKs. This results in the down-regulation of RAS-ERK signalling. In order to evaluate this, we investigated downstream phosphorylation of ERK (pERK) in HEK293T cells transfected with red fluorescence protein-tagged Grb3-3 (RFP-Grb3-3) following incubation with the following extracellular RTK ligands: fibroblast growth factor receptor 2 (FGFR2) ligand fibroblast growth factor (FGF) 9; c-Met receptor ligand hepatocyte growth factor (HGF), and epidermal growth factor receptor (EGFR) ligand interferon-γ (IFNγ). In all three instances, levels of pERK were significantly lower in the presence of exogenous Grb3-3 (Fig. 1a, b).

We have previously provided evidence of an association between Grb2 activation of MAPK pathways and disease progression in colon cancer[10]. We were able to substantiate this observation by demonstrating that the presence of Grb3-3 reduced EGF-induced MAPK signalling indicated by pERK in the human colon adenocarcinoma Caco-2 cell line (Fig. 1c, d). Since this occurred on EGFR activation, the experiment confirms that the influence of Grb3-3 is downstream of the receptor and upstream of RAS.

The MAPK pathway is a key driver of cellular proliferation. We therefore sought to determine the impact of Grb3-3 on this using the 5-bromo-2-deoxyuridine (BrdU) cell proliferation assay. Exogenously expressed Grb3-3 significantly reduced proliferation of both HEK293T (Fig. 1e), and Caco-2 (Fig. 1f) cells. To evaluate whether this reduction in cell proliferation was due to the inability of the corrupted SH2 domain of Grb3-3 to bind to RTK pY-binding sites, proliferation was assayed following the transfection of a Grb2[R82L] mutant, which is characterised by impaired SH2 domain-mediated binding. The presence of Grb2[R82L] reduced HEK293T (Fig. 1e) and Caco-2 (Fig. 1f) cell proliferation, although to a lesser extent than Grb3-3. This suggests that the reduction in cell proliferation that results from an excess of Grb3-3 is due, at least in part, to the defective SH2 domain in Grb3-3.

**Grb3-3 is constitutively associated with Sos.** We sought to characterise the interaction between Grb3-3 and the RAS guanine nucleotide exchange factor Sos. In the first instance, pull-down experiments were performed in HEK293T cells. We used four constructs for these experiments: RFP-tagged full length Grb3-3; two polypeptides containing either the N- or C- terminal SH3 domains along with the SH2 domain including the forty residue deletion corresponding to that found in Grb3-3 (i.e. NSH3SH2$\Delta_{40}$ and SH2$\Delta_{40}$CSH3 respectively), as well as the isolated SH2 domain with the same deletion (SH2$\Delta_{40}$). Grb3-3 was able to immunoprecipitate endogenous Sos under serum-starved and FGF9-stimulated conditions (Fig. 2a). In serum-starved cells Grb3-3 binds to Sos with this interaction mediated by its NSH3 domain. On stimulation with FGF9 the interaction of Grb3-3 appears to be more prevalent via the CSH3 domain (Fig. 2a). There is an apparent reduction in the binding of Grb3-3 on stimulation which is consistent with there being competition between Grb3-3 and Grb2 for binding to Sos. This is likely to be

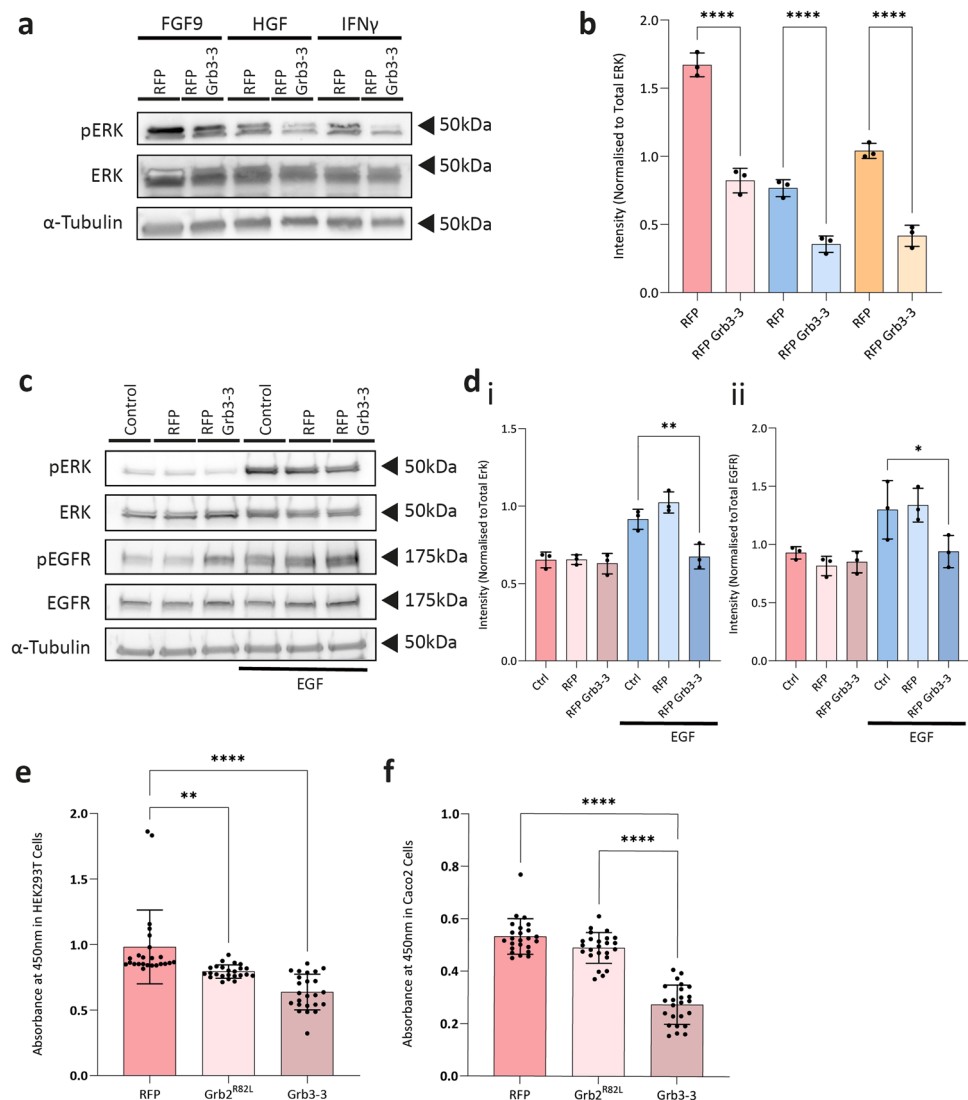

**Fig. 1 Grb3-3 inhibits RAS-ERK signalling and downstream cell proliferation. a, b** A representative western blot and summarised densitometry demonstrating that transfection with exogenous RFP-tagged Grb3-3 significantly reduces phosphorylation of ERK in response to ligand stimulation of receptor tyrosine kinases (RTKs) in HEK293T cells stably transfected with fibroblast growth factor receptor 2 (FGFR2). This experiment demonstrates reduced MAPK activity in the presence of exogenous transfected Grb3-3. α-tubulin is shown as a loading control. See Supplementary Fig. 4 for example uncropped blot. $n = 3$. **c, d** A representative western blot and summarised densitometry demonstrating reduced phosphorylation of ERK (i) in Caco-2 cells in the presence of exogenous transfected Grb3-3, despite persistent phosphorylation of EGFR (ii). α-tubulin is shown as a loading control. See Supplementary Fig. 4 for example uncropped blot. $n = 3$. **e, f** Cell proliferation BrdU incorporation assays demonstrating that in both HEK293T and Caco-2 cells, transfection with a Grb2 mutant characterised by impaired SH2 domain-mediated binding (Grb2$^{R82L}$) or Grb3-3 each resulted in a reduction in proliferation. $n = 24$. $*p \leq 0.05$, $**p \leq 0.01$, $****p \leq 0.0001$.

due to the previous observation that, in the absence of growth factor stimulation, Grb2 has been shown to exist in a stable dimeric form which is incapable of binding to Sos[10,16]. This limits the availability of Grb2 for binding to Sos. Since it is unable to form a dimer interface in the absence of an intact SH2 domain, Grb3-3 is freely available to bind Sos. On stimulation, phosphorylation of Grb2 by FGFR2 dissociates the dimer. The prevailing monomeric Grb2 can compete with Grb3-3 for binding to Sos. These data suggest that under stimulated conditions, which are conducive for Grb2 binding, the NSH3 domains of Grb2 and Grb3-3 compete for binding to Sos. However, in addition Grb3-3 appears to bind via a CSH3-mediated interaction to a different proline-rich sequence on Sos (Fig. 2a).

Additional evidence for the interaction between Grb3-3 and Sos was provided through the observation of fluorescence resonance energy transfer (FRET) between RFP-tagged full length

Grb3-3, NSH3SH2$\Delta_{40}$ or SH2$_{\Delta 40}$CSH3 and GFP-tagged Sos in transfected HEK293T cells (Supplementary Fig. 1a). FRET was observed in both basal and EGF-stimulated conditions, demonstrating that Grb3-3 is constitutively associated with Sos. The FRET data also confirmed that binding to Sos occurs via both the NSH3 and CSH3 domains of Grb3-3.

To determine the site to which Grb3-3 binds Sos, eight glutathione S-transferase (GST)-tagged peptides were synthesised that together incorporate all of the proline-rich motifs within Sos1 (Supplementary Table 1). All but three of the peptides were derived from the C-terminus of Sos; the others were derived from the Sos CDC25 catalytic domain or the pleckstrin homology (PH) domain. Binding of the peptides to purified His-tagged Grb2 and Grb3-3 was assessed by pulldown (Supplementary Fig. 1b). Both Grb2 and Grb3-3 bound to three proline-rich peptides derived from the C-terminus of Sos (namely, peptides Sos-CTail-5, Sos-

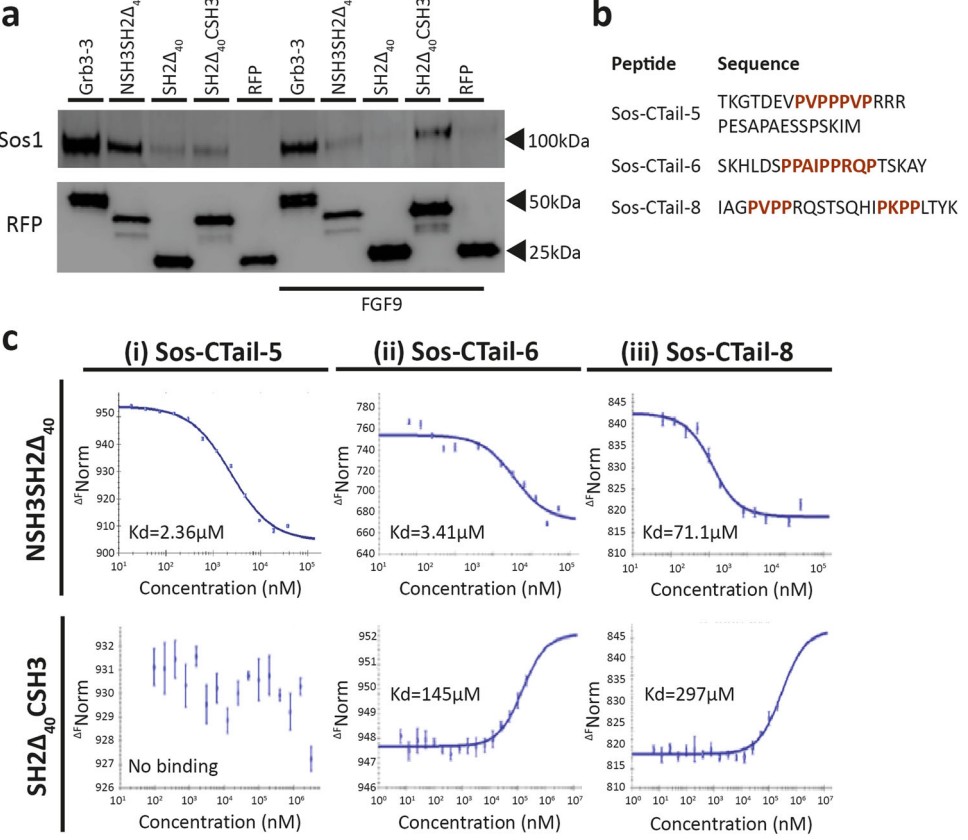

**Fig. 2 Grb3-3 is constitutively associated with Sos and under stimulated conditions competes with Grb2 to bind Sos via its N-terminal SH3 domain.**
**a** Grb3-3 constitutively binds Sos through the N-terminal SH3 domain. A representative western blot from HEK293T cells stably overexpressing fibroblast growth factor receptor 2 (FGFR2) transfected with RFP-tagged full-length Grb3-3, the unique Grb3-3 SH2 domain (SH2$\Delta_{40}$) or truncated Grb3-3 moieties comprising of the Grb3-3 N-terminal SH3 and SH2 domain (NSH3SH2$\Delta_{40}$), the Grb3-3 C-terminal SH3 domain and SH2 domain (SH2$\Delta_{40}$CSH3), in both basal conditions and following FGFR2 stimulation with fibroblast growth factor 9 (FGF9). RFP-tagged proteins were purified and co-immunoprecipitation of endogenous Sos assessed by western blot. $n = 3$. Sos and RFP loading control identified with specific antibody run on same blot: see Supplementary Fig. 5.
**b** In order to identify the site with which Grb3-3 binds Sos, eight proline-rich sequences derived from the C-terminus, CDC25 catalytic domain and the pleckstrin homology (PH) domain of Sos were synthesised (Supplementary Table 1). Binding to His-tagged Grb2 and Grb3-3 was assessed by pulldown assay, with demonstrated the binding of three peptides derived from the c-terminus of Sos (Sos-CTail-5, Sos-CTail-6, Sos-CTail-8) to both Grb2 and Grb3-3 (Supplementary Fig. 1b). The proline-rich regions within these sequences are highlighted in red. **c** MST isotherms for the interactions between three proline-rich peptides derived from the C-terminus of Sos and the Grb3-3 N-terminal SH3 and SH2 domain (NSH3SH2$\Delta_{40}$; top row) and the Grb3-3 C-terminal SH3 and SH2 domain (SH2$\Delta_{40}$CSH3; bottom row). Fitted $K_d$ values are shown individually on each plot and demonstrate that Grb3-3 binding to proline-rich peptides in the C-terminus of Sos was predominantly through the NSH3 domain. $n = 3$ technical replicated and $n = 1$ biological replicate.

CTail-6 and Sos-CTail-8). In addition to the interactions with the C-terminal proline-rich sequences of Sos, only Grb3-3 was also potentially pulled down by peptides Sos-CDC25-2 and 3 derived from the CDC25 domain. These interactions might provide the additional recognition sites for Grb3-3 CSH3 domain highlighted above, being unavailable to Grb2 through conformational restriction not apparent in Grb3-3. This potential selectivity is likely to be linked to recent findings by Kasemein Jasemi et al. highlighting an allosteric mechanism in which the SH2 domain blocks CSH3, ensuring that the interface with NSH3 is the first to form[5].

Microscale thermophoresis (MST) was used to measure the affinity of the interaction between intact Grb2 and Grb3-3 and the proline-rich sequences in peptides derived from the C-terminus of Sos (Supplementary Fig. 1c). Peptides used in the MST experiments were synthesised in truncated form to exclude the GST-tag and residues substantially extending outside the proline-rich motifs (Sos-C-Tail-5, -6 and -8: Fig. 2b). Both Grb2 isoforms bound with similar moderate micromolar affinities in the range typical for proline-rich motif/SH3 domain binding[17]. Comparison of MST-determined affinities for the NSH3SH2$\Delta_{40}$

and SH2$\Delta_{40}$CSH3 constructs of Grb3-3 suggested that binding to proline-rich peptides in the C-terminus of Sos was predominantly through the NSH3 domain (Fig. 2c). These data confirm that under stimulated conditions in cells where both protein isoforms are present, there is likely to be competition between the NSH3 domains of both Grb2 and Grb3-3 for binding to sites on the C-terminus of Sos.

**Competition between Grb3-3 and Grb2 for binding to Sos regulates membrane recruitment.** To demonstrate competition between Grb2 and Grb3-3 for binding to Sos in cells, the effect of an increase in the concentration of transfected *Strep*-tagged Grb3-3 on the ability of transfected RFP-tagged Grb2 to immunoprecipitate endogenous Sos was investigated in HEK293T cells. Overexpression of the full length Grb3-3 construct resulted in a reduction of Grb2-Sos complex formation under both non-stimulated and stimulated conditions (Fig. 3a). The apparent lower level of Grb2-Sos under non-stimulated conditions can be attributed to the prevailing dimeric form of Grb2. Competition between Grb2 and Grb3-3 for binding to Sos was confirmed using

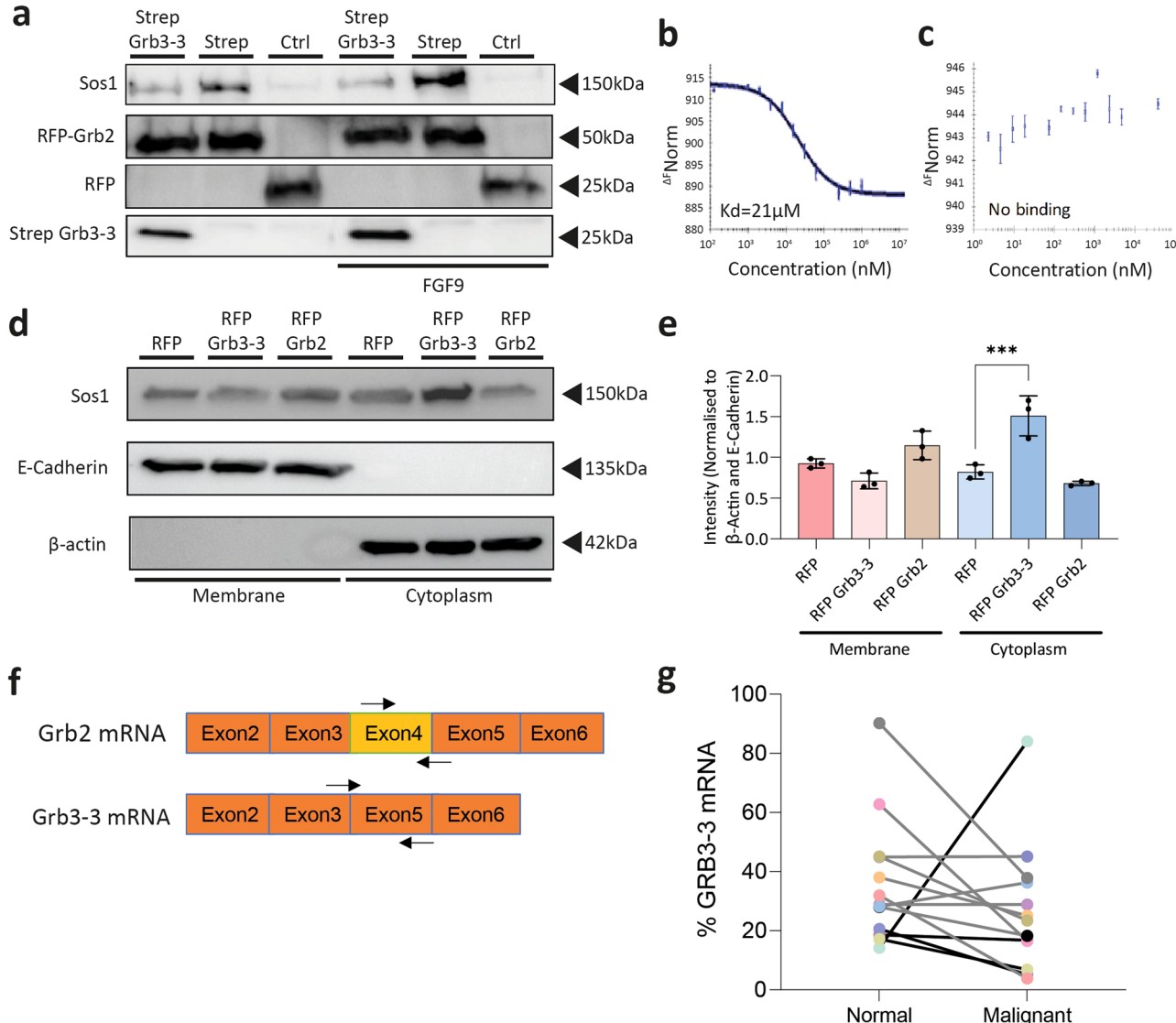

**Fig. 3 Grb3-3 is reduced in colon cancer tissue and inhibits Grb2 binding to Sos, which it sequesters in the cytoplasm. a** A representative western blot showing immunoprecipitation of by RFP-tagged Grb2 of Sos is reduced in the presence of Grb3-3 in HEK293T cells stably expressing FGFR2 in basal conditions and following FGFR2 stimulation by fibroblast growth factor 9 (FGF9). HEK293T cells were transfected with RFP only (Ctrl) or RFP-tagged Grb2 and *Strep*-tagged Grb3-3 (Strep Grb3-3) or *Strep* alone. RFP-tagged proteins were purified and immunoprecipitation of endogenous Sos assessed by Western blot. $n = 3$. RFP-Grb2 and RFP from same blot: See Supplementary Fig. 5. MST measurements confirming competition between Grb2 and Grb3-3 for binding to Sos. Unlabelled Sos C-terminal proline-rich peptide Sos-CTail-5 (10 nM-1 mM) was titrated into a fixed concentration (100 nM) of labelled Grb2 (**b**) in the absence or (**c**) with pre-incubation with 200 μM Grb3-3. $n = 3$ technical replicates and $n = 1$ biological replicate. **d**, **e** A representative western blot and summarising densitometry demonstrating significantly greater Sos cytoplasmic localisation in the presence of exogenous transfected RFP-tagged Grb3-3 (RFP Grb3-3) compared with RFP-tagged Grb2 (RFP Grb2) and RFP control (RFP). Cytoplasmic and plasma membrane fractions were separated by ultracentrifugation followed by immunoblotting for β-actin (cytoplasmic marker), E-cadherin (membrane marker) and Sos. Band intensity was quantified and normalised to β-actin for the cytoplasmic fraction and E-cadherin for the membrane fraction using ImageJ software ($n = 3$; $^*p \leq 0.05$). See Supplementary Fig. 5 for uncropped blots. **f** A schematic representation of the locations for primer annealing (black arrows) used for the splice-sensitive qRT-PCR assay designed to individually detect Grb2 and Grb3-3 transcripts. The Grb2 forward primer annealed to a sequence unique to exon 4, whereas the Grb3-3 forward primer annealed to a sequence spanning the exon 3-exon 5 junction formed by exon skipping. **g** A chart showing the relative concentration of Grb3-3 mRNA as a proportion of total Grb3-3 mRNA and Grb2 mRNA in samples derived from 13 matched colorectal cancer and surrounding normal colonic tissue samples, as determined using a splice-sensitive qRT-PCR assay. $n = 13$.

MST. In the absence of Grb3-3, Grb2 binds to peptide Sos-CTail-5 with an affinity of 21 μM (Fig. 3b). However pre-saturation of the peptide, by incubation with 200μM of Grb3-3, inhibited the ability of Grb2 to bind Sos (Fig. 3c).

Since Grb2 is responsible for translocation of Sos to the plasma membrane through mediating an interaction with an RTK, we next sought to determine the effect of Grb3-3 on Sos cellular localisation. Cell fractionation studies were performed in which

*GRB2* was stably knocked down using shRNA in HEK293T cells. The adaptor protein expression was reconstituted with RFP-tagged Grb2, or RFP-Grb3-3 and the cells were stimulated with FGF9. Ultracentrifugation was used to separate the cytoplasmic fraction (identified with β-actin marker) from the plasma membrane fraction (E-cadherin marker) and levels of endogenous Sos in each fraction were analysed (Fig. 3d, e). Over-expression of Grb3-3 significantly increased localisation of Sos in

the cytoplasm and decreased membrane localisation. Over-expression of Grb2 caused a slight increase in membrane localisation and decrease in cytoplasmic localisation, although this did not reach the predetermined threshold for statistical significance.

Taken together these data are consistent with the binding of Grb3-3 to the C-terminus of Sos functioning to competitively inhibit Grb2 binding to RTKs and the concomitant membrane recruitment of Sos under stimulated conditions. Recruitment of Sos to the plasma membrane is required for RAS activation; thus Grb3-3 inhibits RAS-GTP loading by sequestering Sos in the cytoplasm away from its substrate.

**A reduction in the relative expression of Grb3-3 to Grb2 is seen in normal versus malignant tissue**. We have previously reported a Grb2-mediated regulatory mechanism of MAPK activation in colon cancer[10]. This malignancy is characterised by high MAPK activity, which persists irrespective of RAS mutations[18].Given that we have shown that competition between Grb2 and Grb3-3 for binding to Sos can modulate signalling via RAS, we sought to identify relative Grb3-3 expression in matched malignant and normal colon tissue. It might be expected that lower Grb3-3 expression would result in loss of an inhibitory mechanism to control MAPK activity. Quantitative real-time PCR was used to selectively amplify Grb3-3 and Grb2 in mRNA isolated from 13 patients wild-type for RAS and RAF using forward primers targeted against the unique exon 3-5 junction and exon 4 sequences respectively found in each isoform (Fig. 3f). The specificity of the primers used in this splice-sensitive PCR assay was analysed to ensure there was no cross reactivity between Grb3-3 and Grb2. In addition, sequencing was performed which confirmed the amplicon produced was Grb3-3.

As a proportion of the total expression of both *GRB2* isoforms, nine (69%) of the examined patient samples exhibited relatively lower expression of Grb3-3 in malignant tissue compared to surrounding normal tissue (Fig. 3g). In contrast, a relatively greater expression of Grb3-3 was seen in two of the cancer samples, whilst the levels were broadly similar (within 1%) for two further patients.

**In silico and in vitro analyses identify hnRNPC as a regulator of Grb2 splicing**. Relative differences in the expression of Grb3-3 in colon tissues may occur through altered splicing of exon 4, which results in the expression of Grb3-3 rather than Grb2. Alternative splicing is regulated by splicing factors that bind to regulatory sequences in pre-mRNA, where they function to supress or enhance recruitment of splicing machinery to the nearby 5′ and 3′ splice sites. The two major families of splicing factor proteins are the heterogeneous nuclear ribonucleoproteins (hnRNPs: which typically supress exon inclusion) and the serine arginine-rich proteins (SR proteins; which normally promote exon inclusion).

Potential splicing factors with predicted binding sites within exon 4 were determined in silico using the SpliceAid2 database of human splicing factors expression data and RNA target motifs[19]. Twenty splicing factors were identified as potential binders within exon 4 (Supplementary Table 2). There is significant overlap between predicted splicing factors and their cognate sequences, with most binding sites identified within the first 50 nucleotides of exon 4; in close proximity to the intron-exon boundary (Supplementary Fig. 2).

In order to validate this in silico analysis, an RNA pull-down approach was utilised, in which three 3′-desthiobiotin-labelled RNA oligos, that together spanned the alternative exon and part of the 3′ intron of *GRB2* exon 4, were used (Fig. 4a). Mass

spectrometry (MS) was then used to identify bound proteins. This approach experimentally validated all but three (KHDRBS3, ELAVL2, MBNL1) of the proteins predicted to bind by SpliceAid2 (Supplementary Table 3). The greatest number of matching peptides was observed for hnRNP and SR family members, suggesting these bound with greater affinity and/or were expressed at a higher concentration.

In total, 26 unique hnRNP-family and ten SR family proteins were identified by MS. However, only ten proteins for which binding was identified by MS were predicted to bind by SpliceAid2 analysis (Fig. 4b and Supplementary Table 4), suggesting that other proteins bound indirectly via protein-protein interactions. This is not unlikely given that both hnRNP and SR proteins possess protein-binding domains.

We next sought to identify which of the bound splicing factors plays a functional role in regulating *GRB2* exon 4 inclusion. In a prior analysis, 235 RNA binding proteins (RBPs) in the human erythroleukemic K562 cell line and 237 RBPs in the human liver cancer HepG2 cell line were depleted, generating 221,612 cases of differential splicing identified through whole-transcriptome RNA-sequencing (RNAseq)[20]. This analysis included a majority of the RBPs identified by SpliceAid2 and validated by MS here, though hnRNP G, hnRNP H, ELAVL1 and SRSF10 were not included. We therefore applied replicate multivariate analysis of transcript splicing (rMATS 4.1.0) to replicate control and knockdown data generated in the published study for each of the ten experimentally-validated RBPs[21]. This powerful computational tool has the capacity to assign and identify significant differences in five splicing events; namely skipped exon, alternative 5′ splice site, alternative 3′ splice site, mutually exclusive exons and retained introns. Knockdown of hnRNPC was significantly associated with skipping of exon 4 in K562 cells ($p = 0.009$; Supplementary Table 5). Knockdown of the other identified RBPs was not significantly associated with skipping of *GRB2* exon 4.

**Validation of hnRNPC regulation of *GRB2* exon 4 skipping**. Using SpliceAid2, hnRNPC was predicted to bind to a pentameric polyuracil sequence at position 5-10 of *GRB2* exon 4. In order to confirm the rMATS computational analysis within Caco2 cells, RNA pulldown using a short wild-type (WT) oligonucleotide sequence flanking the 5-10 region was undertaken and compared to a pulldown using a similar nucleotide in which uracil bases were substituted with cytosine. As shown in Fig. 4c, d hnRNPC bound to the WT-sequence with mutation of the WT polyuracil sequence to a polycytosine sequence diminishing but not abrogating binding; suggesting that hnRNPC may bind polycytosine sequences with reduced affinity, or that an additional hnRNPC binding site is present within the 20 nucleotide oligo1. As Grb3-3 expression was shown to be altered in CRC tissue samples, we investigated if hnRNPC was also deregulated in these tissues. hnRNPC expression was significantly diminished in the tumour tissue relative to the surrounding normal colonic tissue (Fig. 4e).

## Discussion
MAPK signalling through RAS plays a fundamental role in the regulation of a wide variety of cellular processes such as proliferation, differentiation, apoptosis, and stress responses. However, aberrant RAS activation through hyperactivity of a RTK and conjunction via a Grb2-Sos complex is a driver of tumorigenesis. Grb3-3 concentration was previously reported to directly impact on EGF-dependent RAS activation[13]. Our work establishes the mechanistic basis for a link between alternative splicing of Grb2 and RAS activation based on the competition between two

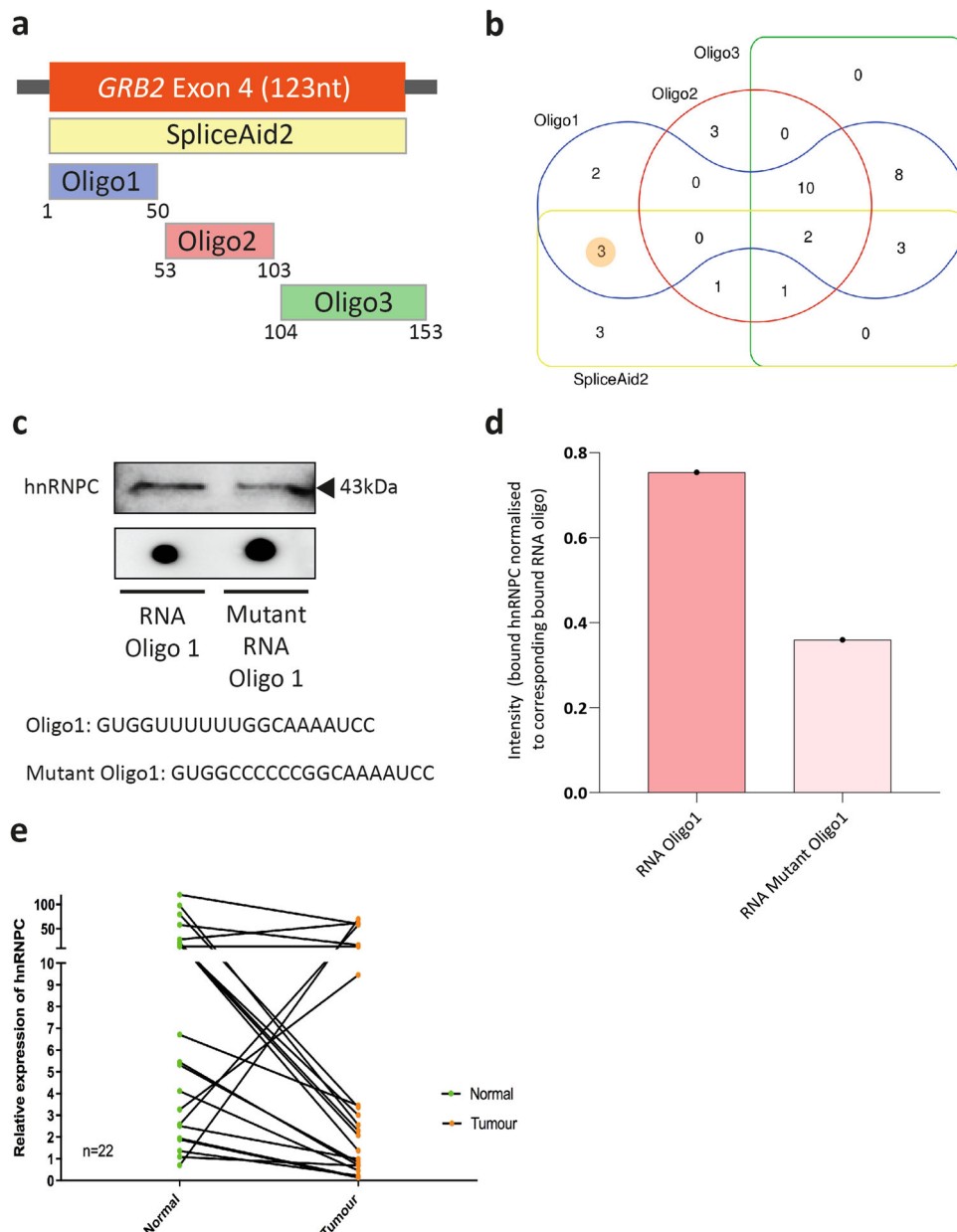

**Fig. 4 hnRNPC regulates Grb3-3 production and is down-regulated in colorectal cancer. a** Schematic overview of the combined coverage of the orthogonal approaches used to identify splicing factors that bind to and therefore regulate inclusion of *GRB2* exon 4. Three desthiobiotin labelled oligos (Oligo1, Oligo2, Oligo3) spanning *GRB2* exon 4 were synthesised and used as bait in RNA pull downs in lysates obtained from HEK293T cells stably transfected with FGFR2. Bound proteins were identified by mass spectrometry. Separately, the bioinformatics programme SpliceAid2 was used to identify splicing factor binding sites in exon 4, in addition to potential cognate splicing factor proteins. **b** A Venn diagram illustrating the number of splicing factor proteins bound to each oligonucleotide, cross-referenced against splicing factors with predicted binding sites in the exon 4 sequence as determined by SpliceAid2. Shaded number corresponds to 3 splicing factor proteins predicted by SpliceAid2 but not validated by MS. $n = 3$. **c, d** Western blot and related densitometry of pulled down samples showing binding of hnRNPC to RNA oligo 1 and reduced binding to mutant RNA oligo 1. $n = 1$. A dot blot demonstrating the relevant labelled RNA oligos used in the pulldown is shown below the western blot bands. See Supplementary Fig. 5 for uncropped blots. **e** Relative mRNA expression of hnRNPC in normal colonic tissue and matched tumour tissue. Green circles represent normal patient colonic tissue, orange circles represent the matched tumour tissue. Black lines connect the patient's normal and tumour tissue samples. $n = 22$.

isoforms, Grb2 and Grb3-3, for binding to the guanine nucleotide exchange factor Sos.

Under stimulated conditions, or conditions where receptor kinase activity is activated through mutation, two protein complexes can form; Grb2-Sos and Grb3-3-Sos, which regulate recruitment of Sos to the plasma membrane. In the presence of a corrupted SH2 domain, Grb3-3 is unable to be recruited to pY sites on RTKs. Thus, when Grb3-3 expression is elevated in cells, the sequestering of Sos1 to the splice variant is inhibitory to RAS

activation. Thus, MAPK pathway signalling is modulated by the relative concentrations of Grb2 and Grb3-3.

These findings provide a mechanistic basis for the action of Grb3-3. Whilst adult tissues are recognised to generally express less Grb3-3 than Grb2, a small number of studies have observed the presence of sufficiently large amounts of Grb3-3 to compete with Grb2, particularly at times characterised by cellular maturation or death. This includes in the developing thymus, and in response to HIV infection in T-lymphocytes[13,22,23]. We extend

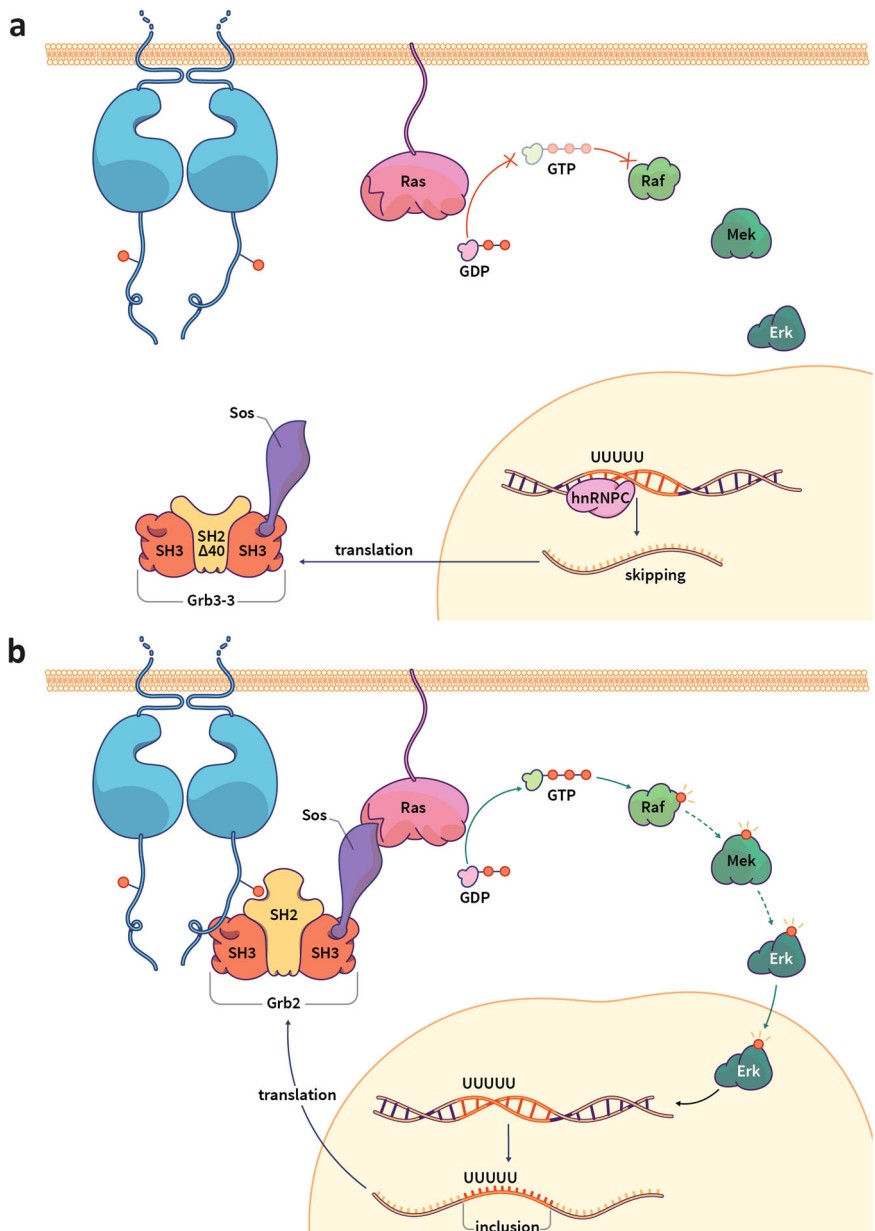

**Fig. 5 Model for molecular mechanism for the control of RAS activation by Grb3-3. a** In Grb3-3 expressing cells, hnRNPC binds to the pentameric poly uracil sequence in *GRB2* exon 4, resulting in the skipping of exon 4 leading to the production of Grb3-3. Grb3-3 binds to the proline-rich C-terminus of Sos and inhibits Grb2 binding to phosphorylated sites (red dots) on tyrosine residues of activated RTKs and concomitant membrane recruitment. Sos remains in the cytoplasm which prevents RAS activation. This ultimately limits ERK translocation to the nucleus and hence cell proliferation. **b** In tumour cells, hnRNPC expression is down-regulated, therefore exon 4 is included producing Grb2. Grb2 binds to Sos and to activated RTKs at the cell surface bringing Sos into close proximity of its substrate RAS. Sos activates downstream signalling via RAS. This results in ERK translocation to the nucleus and transcription of multiple gene leading to cell proliferation.

these findings here by identifying that in colon cancer, which is characterised by evasion of cell death, relative levels of the pro-apoptotic Grb3-3 isoform fall compared with non-malignant tissue. This previously unobserved finding demonstrates a mechanism through which cancer cells may be able to modulate MAPK activity independent of RTK activity; as such, Grb3-3 may act as an inherent cellular control against inappropriate RAS activation and cell proliferation.

In our model (Fig. 5), the prevailing Sos-bound complex is dictated by the relative concentrations of the Grb2 and Grb3-3 splice variants in the cell, for which we identify the splicing factor protein hnRNPC as a potential regulator. In certain tissues in non-malignant cells, where the expression of hnRNPC is high,

inclusion of exon 4 is suppressed resulting in high levels of Grb3-3. Grb3-3 outcompetes Grb2 for binding to the proline-rich C-terminus of Sos and sequesters Sos in the cytoplasm away from its substrate RAS. This inhibits activation of the RAS-ERK MAPK signalling cascade and proliferative outcome (Fig. 5a). In tumour cells, expression of hnRNPC is reduced, therefore exon 4 is included resulting in elevated levels of Grb2 and reduced expression of the negative regulator Grb3-3. Grb2 outcompetes Grb3-3 for binding to Sos which mediates membrane localisation and activation of the RAS-ERK signalling cascade (Fig. 5b). This model must however be caveated by the lack of evidence shown here for a direct correlation between hnRNPC expression and corresponding Grb3-3 expression.

The link between alternative splicing and cancer is well established[24,25]. Changes in alternative splicing events have been extensively reported to facilitate acquisition of the hallmarks of cancer. A role of hnRNPC in regulating cell proliferation and differentiation has also previously been reported[26,27], however only a handful of hnRNPC-regulated alternative splicing events have been described[28,29]. Interestingly, there is evidence that hnRNPC interacts with both Grb2 and Grb3-3[14]. Interaction of the former with hnRNPC is inhibited by poly(U) RNA, whereas this enhances its interaction with Grb3-3. This, and the findings presented here, suggest that these interactions fulfil different biological functions; potentially giving rise to a feedback loop through which a stable relative concentration of Grb2 and Grb3-3 can be maintained. This feedback might have particular importance in tissue where controlled cell proliferation is physiologically relevant such as renewal of intestinal lining in colon crypts.

With drug resistance in cancer therapy remaining a major limiting factor and exhaustive pharmaceutical efforts to inhibit RAS activity largely resulting in failure, novel approaches to inhibition of RTK-derived signalling are urgently sought. This work underlines the importance of the interaction of the SH2 domain of Grb2 to enabling sequestered Sos to be delivered to an RTK and the plasma membrane-localised RAS. This appears to be controlled by a feedback loop involving Grb3-3. Several attempts to abrogate the interaction of the Grb2 SH2 domain with RTKs using small molecule inhibitors have been reported[30–32] however these, predominantly peptidomimetic molecules, have not shown efficacy in the clinic. Our data indicate a need to better understand the role that perturbation of the Grb2:Grb3-3 ratio, and in particular the specific role of Grb3-3, play in establishing Ras signalling in disease states such as cancer.

## Methods

**Materials & reagents**. Recombinant human fibroblast growth factor 9 (FGF9; #273-F9-025), hepatocyte growth factor (HGF; 294-HG), transforming growth factor alpha (TGFα; 239-A) and epidermal growth factor (EGF; 236-EG) were purchased from R&D systems. Antibodies raised against phospho-p44/42 MAPK (ERK1/2) (Thr202/Tyr204) (#9101), p44/42 MAPK (ERK1/2) (#9102), α-Tubulin (11H10) (#2125), phospho-EGF receptor (Tyr1068) (D7A5) (#3777), EGF receptor (D38B1) XP® rabbit mAb 1:1000 (#4267), GRB2 antibody (#3972), E-Cadherin (24E10) (#3195), β-Actin (13E5) (#4970), hnRNP A0 (#4046), RAS (27H5) (#3339), and GST (26H1) (#2624), (#7076) were purchased from Cell Signalling, as was an anti-rabbit IgG HRP-linked antibody (#7074). An anti-RFP rabbit antibody (ab62341) was purchased from Abcam and an anti-SOS1 mouse antibody (WH0006654M1) purchased from Sigma-Aldrich. DH5α competent and One Shot™ E. coli were purchased from Thermo Fisher Scientific. BL21 (DE3) competent E. Coli were purchased from NEB.

**Mammalian cell culture**. Mammalian cells were maintained as sub-confluent cultures at 37 °C with 5% carbon dioxide (CO₂). When not passaged, 50% media changes were undertaken at two-three day intervals. Human colorectal adenocarcinoma Caco-2 (HTB-37™) cells, the highly-transfectable human embryonic kidney (HEK293T; CRL-3216™) cell line and the human cervix adenocarcinoma HeLa (CCL-2™) cell line were purchased from American Type Culture Collection (ATCC; Virginia, USA). The HEK293T line was stably transfected with FGFR2, as has been previously described[33]. This cell line has been utilized in multiple previous studies in order to enable the study of the effect of FGFR2 up-regulation alongside other endogenously expressed RTKs[16,33]. All cells were tested for, and found to be free of, mycoplasma contamination using the LookOut Mycoplasma Detection kit (Thermo Fisher Scientific) applied at monthly intervals. Caco-2 cells were cultured in Eagle's Minimum Essential Medium (EMEM; Sigma Aldrich) supplemented with 10% (v/v) fetal bovine serum (FBS; Sigma Aldrich) and 50 μg/ml gentamicin (Sigma Aldrich). Modified HEK293T and HeLa cells were cultured in Dulbecco's Modified Eagle Medium (DMEM) supplemented with 10% (v/v) FBS, 50 μg/ml gentamicin and 7 μg/ml puromycin (Sigma Aldrich) to maintain knockdown of Grb2 or stable expression of FGFR2. RIPA Lysis and Extraction Buffer (#89900) was obtained from Thermo Fisher Scientific. Bovine serum albumin (BSA; A9647) was purchased from Merck.

**Plasmids**. Recombinant plasmids for bacterial and mammalian protein expression were produced by molecular cloning. Genes of interest were amplified using polymerase chain reaction (PCR) and, following dedicated restriction enzyme digests, DNA ligations were established in 10 μl volumes with a 3:1 ratio of insert:vector DNA using T4 DNA ligase (NEB #M0202) and DNA ligation buffer (NEB #M2200). The His-tagged pET28a bacterial expression vector was purchased from Novagen and used to separately express full length His-tagged Grb2 [pET28a-Grb2] and both full-length [pET28a-Grb3-3] and truncated [pET28a-Grb3-3-NSH3SH2Δ₄₀, pET28a-SH2, pET28a-SH2Δ₄₀CSH3] His-tagged Grb3-3. A separate pET28a bacterial expression vector within which residues 564-1059 of the catalytic domain of Sos are cloned [pET28a-Soscat] was a kind gift from Professor Alex Breeze, University of Leeds. The pGEX4T bacterial expression vector was purchased from GE and used to express sequences incorporating different proline rich motifs found within Sos [pGEX2T-Sos1, pGEX2T-Sos2, pGEX2T-Sos3, pGEX2T-Sos4, pGEX2T-Sos5]. The pcDNA 3.1 vector was purchased from Addgene and was used to express full length RFP-fused Grb2 [pcDNA3.1-Grb2], both full length [pcDNA3.1-Grb3-3] and truncated [pcDNA3.1-Grb3-3-NSH3SH2Δ₄₀, pcDNA3.1-Grb3-3-SH2, pcDNA3.1-Grb3-3-SH2Δ₄₀CSH3] RFP-fused GRB3-3, and both myc-[pcDNA3.1-hnRNPC-Myc] and RFP-[pcDNA3.1-hnRNPC-RFP] tagged hnRNPC. For cell-base Strep-tag pull-down experiments, the Grb2 gene was amplified with an N-terminal Strep-tag and cloned in a pcDNA3.1 vector [pcDNA3.1-strepGrb2] Mutagenesis (XX) was carried out using the Q5® site-directed mutagenesis kit (E-0554S) New England BioLabs) in accordance with the manufacturer's protocol. As detailed below, the pET30-2-GAPDH bacterial expression vector was purchased from Addgene and used to produce the standard curve for the splice-sensitive PCR assay by an alternative process of TA cloning. For cell based Strep-tag pull-down experiments, full-length Grb2 and Grb3-3 were amplified with an N-terminal Strep-tag and cloned in a pCNDA-Myc vector [pcDNA-Myc-Strep-Grb2, pcDNA-Myc-Strep-Grb3-3].

**Transformation, protein expression and protein purification**. Recombinant DNA plasmids produced through molecular cloning were transformed into DH5α competent cells for DNA production. Briefly, between 1–100 ng plasmid DNA was incubated on ice with 50 μl competent cells for 30 min prior to a 45 min period of heat shock at 42 °C. Cells were recovered by addition of 950 μl SOC media and incubated at 250 rpm with constant agitation at 250 rpm for 1 h, then pelleted by centrifugation at 5000 rpm for 30 s. Once the supernatant was removed, cells were re-suspended in 50 μl SOC media and grown on agar plates inoculated with an appropriate selection antibiotic.

Proteins were expressed in BL21 (DE3) cells. Following an overnight culture, 20 ml BL21 cells were used to inoculate 1 litre LB media conditioned with appropriate antibiotic (50 μg/ml kanamycin for plasmids with a pET28a backbone, and 50 μg/ml ampicillin for plasmids with a pGEX4T1 and a pET30-2-GAPDH backbone. Bacterial cultures were subsequently incubated at 37 °C with constant agitation (200 rpm) until an OD₆₀₀ of 0.7 was reached. The culture was then cooled to 18 °C and 0.5 mM isopropyl β-D-1-thiogalactopyranoside (IPTG) added for 12 hours in order to induce protein expression. Once harvested, cells were suspended in Talon buffer A (a pH 8.0 solution of 200 mM Tris, 150 mM NaCl and 1 mM β-mercaptoethanol) and lysed by sonication. Cellular debris was removed by centrifugation at 20,000 rpm for 1 h at 4 °C and proteins then purified from the soluble fraction using an ÄKTA Pure protein purification system. Proteins were then eluted using Talon buffer B (a pH 8.0 solution of 20 mM Tris, 150 mM NaCl, 200 mM imidazole and 1 mM β-mercaptoethanol), concentrated to 2 ml and applied to a Superdex SD75 column within HEPES buffer (a pH 7.5 solution of 20 mM HEPES, 150 mM NaCl and 1 mM TCEP). All proteins were assessed via SDS-PAGE in order to ensure adequate purity. All plasmids have been deposited on the Addgene Database and Accession Numbers are included in Supplementary Data 2.

**Mammalian cell transfection**. Mammalian cells were transfected with plasmid DNA using either TransfeX Reagent (ATCC) or Metafectene (Biontex). Briefly, 24 h prior to transfection cells were seeded at 50–60% confluency and maintained in proliferative media at 37 °C with 5% CO₂. Fresh proliferative media was added immediately prior to transfection. In accordance with manufacturer instructions, transfection complexes were first prepared through the addition of 5 μl TransfeX Reagent or Metafectene to 100 μl Opti-MEM 1 Reduced-Serum Medium respectively containing 2.5 μg or 2 μg plasmid DNA for each experimental well. Following a 15 min incubation period, pre-warmed transfection complexes were added to cells, with the culture vessels rocked in order to aid their distribution. Each transfected culture was then incubated for at least 24 h before media change or experimentation. Where two plasmids were co-transfected, the volume of metafectene used was doubled.

**Ligand stimulation of mammalian cells**. Cells were maintained at 80% confluency prior to ligand stimulation. Twenty-four hours prior to stimulation, cells were washed three times in PBS and then serum-starved through culture in media not supplemented with FBS. Following this cells were stimulated with 10 ng/ml fibroblast growth factor 9 (FGF9), hepatocyte growth factor (HGF), interferon-ɣ (IFNɣ), epidermal growth factor (EGF) or transforming growth factor-α (TGFα). Cells were incubated with FGF9 for 15 min, and with all other ligands for 5 min, prior to lysis.

**Subcellular fractionation**. Cell pellets were prepared by centrifugation at 13,000 $g$ for 20 min prior to cell fractionation studies. Each pellet was then re-suspended in 1 ml homogenisation buffer (250 mM sucrose, 3 mM imidazole pH 7.4, 1 mM EDTA, protease inhibitor cocktail) and passed through a 21 gauge needle ten times, then a 27 gauge needle ten times. Following a 20 min incubation on ice, samples were centrifuged at 600 $g$ for 5 min, enabling separation of a nuclear pellet from the remainder of the cellular fraction. The separated supernatant was centrifuged again at 20,000 $g$ for 30 minutes in order to pellet mitochondria. The resultant cytoplasm and membrane containing supernatant was again centrifuged, this time at 10,000 g for 60 min, in order to separate the two components. Microsomal membranes were then applied to a 40–100% sucrose gradient and centrifuged at 10,000 $g$ in order to remove the golgi apparatus and endoplasmic reticulum. The plasma membrane was obtained by ultracentrifugation at 100,000 $g$ for 1 h, which permits removal of golgi apparatus and endoplasmic reticulum. The plasma membrane was then re-suspended in RIPA Lysis and Extraction buffer.

**Western blotting**. Following experimental, cell lysates were obtained in RIPA Lysis & Extraction Buffer. Protein concentration was measured using a colorimetric Pierce™ BCA Protein Assay kit (#23225, Thermo Fisher Scientific). Dependent on the target epitope, between 20–40 μg protein lysate was resolved by sodium dodecyl sulfate-polyacrylamide gel electrophoresis (SDS-PAGE) and wet transferred to a polyvinylidene fluoride (PVDF) membrane. Membranes were blocked using either 5% milk or, if the membrane was to be used for the detection of a phosphorylated epitope, 5% BSA. After blocking, membranes were incubated overnight at 4 °C in primary antibody, resuspended at a 1:1000 dilution in either 5% milk or 5% BSA. Membranes were subsequently washed in Tris-Buffered Saline (Tris 20 mM, NaCl 150 mM) with 0.1% Tween® 20 detergent (TBST) and incubated for an hour at room temperature in horseradish peroxidase (HRP)-linked secondary antibody, used at a 1:2500 dilution. A second wash step was then undertaken in TBST and Pierce™ Western Blotting Substrate (#32106, Thermo Fisher Scientific) used as a HRP chemiluminescence substate. Signal detection was carried out using a Syngene G:BOX. Relative band intensity was quantified by densitometry using ImageJ, with all target bands corrected for background.

**RNA extraction**. RNA was extracted from formalin-fixed, paraffin embedded tissue sections using RNeasy® FFPE kit (#73504, Qiagen) in accordance with manufacturer instructions. Six 5 μM thick tissue sections were used for each extraction, each of which was first deparaffinised through the addition of Deparaffinisation Solution (#19093, Qiagen). Following RNA extraction, sample quality was assured through assessment of 260 nm versus 280 nm and 230 nm versus 260 nm ratios using a NanoDrop 2000 spectrophotometer (Thermo Fisher Scientific).Only samples with a 260/280 ratio of 1.7–2.0 and a 230/260 ratio of 2.0–2.2 were used in subsequent assays. RNA integrity was subsequently confirmed by analysis on a denaturing agarose gel.

**Complementary DNA (cDNA) synthesis**. cDNA was transcribed from RNA for gene expression analyses. For each reaction, 1.5 μg RNA was transcribed in a total reaction volume of 20 μl using a final concentration of 2.5 μM Oligo dT primer (#18418020, New England BioLabs) and 10 mM dNTP. Reactions then proceeded via incubation at 65 °C for five minutes, followed by the addition of 1 μl Super-Script® IV Reverse Transcriptase (Thermo Fisher Scientific) and appropriate buffers, in line with manufacturer guidance.

**Splice-sensitive quantitative polymerase chain reaction (qRT-PCR) assay**. A splice-sensitive qRT-PCR assay was developed in order to allow for discrimination between the mRNA levels of Grb2 and Grb3-3. This took advantage of the omission of exon 4 to generate Grb3-3. Specifically, a primer pair was designed in which the forward primer targeted a region unique to exon 4, thereby permitting the detection of Grb2 alone with an expected amplicon size of 76 base pairs (forward: 5′-CCCAAGAACTACATAGAAATGAAACC-3′, reverse: 5′-CCGCTGTTTGCTAAGCATTT-3′). Conversely, in order to detect Grb3-3 alone, a primer pair was generated in which the forward primer spanned the unique exon 3 – exon 5 junction generated from exon 4 skipping, with an expected amplicon size of 223 base pairs (forward: 5′-AACCACATCCGTTTGGAAAC-3′, reverse: 5′-TTCTGGGGATCAAAGTCAAA-3'). Separate primers were designed against the housekeeping GAPDH gene (forward: 5′-AGAAGGCTGGGGCTCATTTG-3′, reverse: 5′-TTCTCATGGTTCACACCCATG-3′).

All primers were used for qRT-PCR at a final concentration of 250 nM in a reaction mix comprising of 100 ng cDNA template and SensiMix™ SYBR®. Each biological replicate was analysed in duplicate and qRT-PCR carried out within a Rotor-Gene Q (Qiagen). The applied thermal conditions included a 10 min 95 °C polymerase activation step followed by 40 cycles of 95 °C for 15 s, 60 °C for 30 s and 72 °C for 30 s whilst reading green fluorescence. Concentrations of Grb2 and Grb3-3 were then determined using standard curves.

Prior to experimentation, all primers were confirmed to deliver an acceptable reaction efficiency (Grb3-3: 105%, Grb2: 101%, GAPDH: 106%). Specificity of the Grb3-3 primer pair was confirmed by transfection in to HeLa cells of varying concentrations of Grb2 and Grb3-3, prior to RNA isolation,cDNA synthesis and qRT-PCR. As summarised in Supplementary Fig. 3a, gel electrophoresis of the PCR

products resulting from the use of Grb3-3 primers showed that only a single band was produced, and this corresponded to the expected amplicon size of between 200–300 base pairs. This was seen in all samples, including those with no exogenous Grb3-3 transfection, indicating that the assay was sufficiently sensitive to detect endogenous Grb3-3 concentration. Similarly, as shown in Supplementary Fig. 3b, use of the Grb2 primer pair resulted in the production by qRT-PCR of a single product measuring as expected less than 100 base pairs in length, even in the presence of exogenous (5 μg) transfected Grb3-3 cDNA.

In order to further validate the specificity of the Grb3-3 primers, the PCR fragment from sample 9 shown in Supplementary Fig. 3a was gel-extracted and cloned into a TA vector. The cloned sequence was confirmed by sequencing to match that of Grb3-3.

**Agarose gel extraction**. RNA bands of interest were extracted from agarose gel using the QIAquick Gel Extraction Kit (#28706, Qiagen), in accordance with manufacturer instructions. DNA was eluted in 50 μl Buffer EB.

**TA cloning**. A TA cloning kit (K2070-20, K2070-40, Life Technologies) was used in order to validate by sequencing the products generated by the splice-sensitive PCR assay. Briefly, and as per manufacturer instructions, ligation of gel-extracted PCR product was established in 10 μl volumes with a 2:1 ratio of insert: vector DNA, using Invitrogen™ ExpressLink™ T4 DNA ligase (#11370862, Thermo Fisher Scientific) and Introgen™ ExpressLink™ T4 DNA ligase buffer (#11370862, Thermo Fisher Scientific). Ligated plasmids were subsequently transformed into Invitrogen™ One Shot™ Top10 competent cells (#10358022, Thermo Fisher Scientific), as for DH5α competent cells, and plated onto LB agar plates containing 50 mg/ml kanamycin, 40 mg/ml X-Gal and 100 mM IPTG. Successful TA cloning results in the formation of white, as opposed to blue, colonies; with the former then selected for sequencing.

**Glutathione-S-transferase (GST) pull-down**. GST pull downs were performed as has previously been described[34]. Briefly, GST fusion constructs were immobilised on glutathione cross-linked agarose beads. 20 μl of GST-fusion protein bead suspension was then added to 200 μl cell lysate or purified protein and incubated overnight at 4 °C on an orbital rotator. Beads were then pelleted by centrifugation at 1000 rpm for 1 min and washed in PBS-T. This wash step was repeated five times. Bound proteins were eluted in Laemmli buffer (#1610737, BioRad Laboratories) and analysed by SDS-PAGE.

**Red fluorescent protein (RFP) pull-down**. Mammalian cells were transfected as outlined above with RFP-tagged fusion proteins and lysates prepared in RIPA Cell Lysis Buffer following experimentation. RFP-tagged proteins were then precipitated within lysates through a one hour incubation at 4 °C with RFP-Trap® Agarose. Beads were subsequently pelleted by centrifugation at 1500 rpm for one minute and then washed in PBS-T. This wash step was repeated five times. Bound proteins were then analysed by western blotting.

**RNA labelling**. Three 50 oligonucleotide sequences that together spanned Grb2 exon 4 and part of the 3′ intron were purchased from Integrated DNA Technologies. (Fig. 4a). The sequences of these oligopeptides were as follows: RNA Oligo1: GTGGTTTTTTGGCAAAATCCCCAGAGCCAAGGCAGAAGAAATGCTTAG CA. RNA Oligo2: AGCGGCACGATGGGGCCTTTCTTATCCGAGAGAGT GAGA GCGCTCCTGGG. RNA Oligo3: GACTTCTCCCTCTCTGTCAAGT AAGTATTT CCTGCTGCAGTTGCCTGGAA. Each RNA oligonucleotide was labelled at the 3′ end with a single biotinylated cytidine (bis)phosphate using the Pierce™ RNA 3′ End Desthiobiotinylation Kit (#20163, Thermo Fisher Scientific). In accordance with manufacturer instructions, the ligase reaction proceeded overnight at 16 °C and comprised of 50pmol of the RNA oligonucleotide, 1nmol biotinylated cytidine bisphosphate, 40 units T4 RNA ligase, 15% (v/v) polyethylene glycol, 40 units RNase inhibitor and RNA ligase reaction buffer. The RNA ligase was subsequently extracted through the addition of choroform isoamyl alcohol and RNA precipitated from the aqueous phase through the addition of 250 mM NaCl, glycogen and ethanol for a period of 1 h at −20 °C.

**RNA pull-down**. Labelled RNA oligonucleotides were immobilised on to streptavidin beads using the Pierce™ Magnetic RNA-Protein Pull-Down Kit (#20164, Thermo Fisher Scientific). For RNA immobilisation, 50 pmol labelled RNA oligonucleotide was added to the beads and incubated at room temperature in the presence of RNA Capture Buffer for 30 min. Following a wash step, RNA-protein binding continued via the addition for one hour at 4 °C of 60 μl mammalian cell lysate to 50pmol labelled RNA in the presence of protein-RNA binding buffer and glycerol.

**Mass spectrometry**. Proteins/peptides were extracted from gels using standard protocols to avoid contamination. Dissolved proteins were run on S-trap micro columns (Protifi) to denature and purify and digested with trypsin. Proteins/ peptides were analysed on a Bruker Daltonics UltrafleXtreme 2 mass spectrometer (Bruker) by matrix-assisted laser desorption/ionization-tandem mass spectrometry

(MALDI-MS/MS). Reads were searched for peptide modifications against the human Uniprot database. All raw mass spectral data is included in Supplementary Data 1.

**Cell proliferation assay**. Cell proliferation was assayed by enzyme-linked immunosorbent assay (ELISA) using bromodeoxyuridine/5-bromo-2′-deoxyuridine (BrdU) labelling in accordance with manufacturer instructions (ab126556, Abcam). Cells were seeded at a density of 10,000 cells per well in a 96 well plate and incubated with BrdU for a period of 24 h following experimentation. Absorbance was then read at room temperature at 420 nm using a BioTek™ PowerWave™ Microplate Spectrophotometer.

**Confocal microscopy and fluorescence resonance energy transfer (FRET)**. HEK293T cells co-transfected with RFP-tagged full length Grb3-3, NSH3SH2$\Delta_{40}$ or SH2$\Delta_{40}$CSH3, and GFP-tagged Sos, were cultured on glass coverslips for 24 h before experimentation. As controls, cells expressing RFP- and GFP- tagged vector alone were also analysed. Following a period of serum-starvation or ligand stimulation, cells were fixed by addition of 4% (w/v) paraformaldehyde at pH 8.0 for 20 min, washed and then mounted onto a slide with mounting medium containing DAPI. Imaging was performed using a Zeiss LSM880 confocal multiphoton laser scanning microscope following excitation at 405 nm for DAPI was excited at 405 nm, 488 nm for GFP, and 561 nm for RFP. For FRET analyses, GFP was excited at 488 nm and RFP emission was detected.

**Microscale thermophoresis (MST)**. Equilibrium dissociation constants ($K_d$'s: binding affinities) were measured using the Monolith NT.115 (NanoTemper Technologies, GmbH). Proteins were fluorescently labelled with Atto488 NHS ester (Sigma Aldrich) according to the manufacturer's protocol. Labelling efficiency was determined to be 1:1 (protein:dye) by measuring the absorbance at 280 and 488 nm. MST experiments were set up using 16 samples, all of which contained the fluorescently labelled protein at a final concentration of 100 nM. The unlabelled protein was added to all but one of the samples and serially diluted by 2 fold. The samples were then transferred into glass capillaries. Measurements were performed at 25 °C in a buffer containing 20 mM HEPES, 150 mM NaCl, 1 mM DTT and 0.01% Tween 20 at pH7.5. Data analysis was performed using Nanotemper Analysis software, v.1.2.101 and was plotted using Origin 7.0. All measurements were conducted as triplicates and the error bars were presented as the standard deviations of the triplicates.

**SpliceAid2**. The SpliceAid2 database was used to identify potential splice site sequences within exon 4 of the *GRB2* gene, as well as their cognate splicing factors[19]. The following *GRB2* exon 4 sequence was used as input for this analysis: GUGGUUUUUUGGCAAAAUCCCCUGUGCCUUGGCAGAAGAAAUGCUUAGCAAACUGCGGCUCGAUGGGGCCUUUCUUAUCCGAGAGAGUGAGAGCGCUCCUGGGGACUUCUCCCCUCUCUGUCAA.

**Multi-variant analysis of transcript splicing (rMATS)**. The robust statistical rMATs model was used to detect differential alternative splicing of Grb2 from replicate RNA-Seq data generated by Van Nostrand et al.[20].

**Statistics and reproducibility**. For all experiments, data were analysed using ANOVA (GraphPad, Prism 9). $P$-values $\leq 0.05$ were considered statistically significant.

For all statistically quoted values $n \geq 3$ biological replicates.

**Reporting summary**. Further information on research design is available in the Nature Research Reporting Summary linked to this article.

## Data availability

All data required to evaluate the conclusions of this study are included within the main report and supplementary materials. The complete mass spectral raw data for oligos 1, 2, and 3 are provided in Supplementary Data 1. Plasmids have been deposited in the Addgene Repository and the Accession ID Numbers are included in the table in Supplementary Data 2. These Accession Numbers are as follows: pET28a-SH2 – 190908; pET28a-SH2Δ40CSH3 – 190907; pET28a-NSH3SH2Δ40 – 190906; pET28a-Grb3-3 – 190905; pET28a-Grb2 – 190904. Source data for the following: Fig. 1b; Fig. 1d.i, 1d.ii; Fig. 1e; Fig. 1f; Fig. 3e; Fig. 3g; Fig. 4d; Fig. 4e are provided in Supplementary Data 3. RNA oligo sequences are included in Methods section under RNA Labelling. Uncropped blots are provided in Supplementary Figs. 4 and 5.

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

## Acknowledgements
This work was funded by CRUK grant C57233/A22356 awarded to J.E.L. C.S. and A.K.S. were supported by Wellcome Trust funded Ph.D. Studentships.

## Author contributions
Experiments devised by C.S., P.Q. and J.E.L. Experiments performed by C.S., A.K.S., S.K. and K.M. Manuscript written by C.S., A.K.S., C.M.J. and J.E.L. Figures produced by C.S., A.K.S., S.K. and C.M.J.

## Competing interests
The authors declare no competing interests.

## Ethics
The use of patient samples was approved by the National Research Ethics Service (London – Bloomsbury Research Ethics Committee; 12/LO/1217). Patient consent for use of data is included in this ethics approval.
