## [Peer Review File · Communications Biology]

Reviewers' comments:

Reviewer #1 (Remarks to the Author):

This is a nice study that demonstrates that expression of a Grb2 splice variant Grb3-3 results in decreased activation of the Ras-MAPK pathway and that this is due competition with Grb2 for SOS binding resulting in a cytoplasmic complex sequestered away from being able to activate Ras. The potential clinical relevance of this splice variant is demonstrated by seeing reduced Grb3-3 levels in tumour vs normal tissue ie. suggestive of a mechanism for enhancing Ras pathway activity in tumours that are otherwise wild-type for Ras. A screen for mediators of splicing identified hnRNPC and this gene is downregulated in tumour vs associated normal tissue.

The first area of clear novelty in this study is the idea of and evidence for competition between Grb3-3 vs Grb2 for SOS - it provides a mechanistic basis for previous observations that are in essence the same as those here showing downregulation of Ras-MAPK signalling. NB. in the 1994 Fath et al Science paper where the effects on RTK-Ras signalling were first observed they relied on functional assays rather than the precise immuno-reagents that are now available but the conclusions are the same. This earlier paper also identified the effect on SH2-mediated interactions and speculated about effects on SOS regulation although the mechanism was not defined. The profiling of tumour vs normal is also an advance on this previous work although it should be noted this is circumstantial evidence - it is not definitively demonstrated in the clinical samples that there are differences in Ras-MAPK output that correlate with Grb3-3 expression. The second half of the paper involves a well conducted screen for splicing mediators responsible for Grb3-3 expression. hnRNPC is the lead candidate that is shown to bind to a relevant Grb2 oligo although I note the absence of formal validation that manipulations of hnRNPC expression/function result in concomitant changes in Grb3-3 expression. hnRNPC has previously been shown to bind to Grb2 and Grb3-3 proteins (Romero et al 1998) and so a comment on this in the discussion would be beneficial ie. could this be a mechanism for positive/negative regulation?

In summary the work is nicely performed (inc. appropriate stats analysis). I am convinced by the central observations, they represent an advance on previous work by providing both mechanistic insight and potential clinical relevance.

Edits/further work

I'm satisfied with the paper as it stands although if the clinical sample correlations and/or the hnRNPC manipulations could be performed they would enhance confidence in the observations. Discussion of hnRNPC:Grb2/Grb3-3 protein would be desirable. There was a mix up with labelling of Figure 3 panels in the text - Figure 3ef are Figure 3de etc.

Reviewer #2 (Remarks to the Author):

The Grb2 splice variant, Grb3-3, is a negative regulator of RAS activation.

Ketchen et al.

In this manuscript the authors study the role of a splicing variant of the adaptor protein GRB2 in the regulation of the activation of RAS proteins downstream of RTK. The authors show that GRB3-3 can bind the RAS GEF SOS independent of RTK activation. This binding prevents the binding of SOS to RAS due to sequestration of SOS in the cytoplasm. They also show results indicating that GRB2 and GRB3-3 compete for the interaction with SOS. The authors present evidence that GRB3-3 is a putative tumour suppressor which RNA expression is regulated by hnRNP-dependent splicing. The data presented helps expanding the knowledge on the role of GRB2 in RAS/MAPK regulation and in particular sheds light about the role of the isoform GRB3-3 in cancer. However, while the working model seems to be supported by some of the data presented here, there are some problems with the way the data is presented, the description of the experimental approaches and some further experiments/analysis would increase the support for the author's conclusions. Thus, while the work presented here might be of interest for the readership of Communications biology but improvements are necessary before it can be recommended for publication.

Major points:

1. A big problem with the current manuscript is the lack of sufficient description of the material and methods, and lack of information in the figure legends. This complicates the interpretation of some of the results (see below comment about figure 4D). In general, there are no references for the protocols used, there is no description of critical reagents such as lysis and washing buffers, the plasmids used and some steps of the protocols necessary to allow the reproducibility of the data by the community. The description of the MS analysis set up is especially insufficient, and there is not indication if the results have been deposited in any public repository. This has to be corrected.
2. Does GRB3-3 regulate EKR levels in the stable cell lines? In figure 1A and E there seems to be a difference on the level of expression of these kinases in the two cell lines. It would be better if the authors showed the levels of EKR activation in figure 1A in the absence of stimuli to know the basal levels in the stable cell lines. Additionally, do the authors have any explanation why HGF does regulate levels of ERK expression in figure 1A?
3. Figure 1C, the pull down should be quantified against the total levels of RAS not GST-RBD. This is the common way of presenting this kind of experiments. It is also common to present this as % of activation (i.e <https://www.sciencedirect.com/science/article/pii/S2451945617303239#fig4>). The inclusion of a blot with RAS expression levels in cell extract would also increase confidence that no changes shown are due to a possible regulation of RAS level by expression of GRB3-3
4. Figure 1G and H, the labels should indicate that the GRB2 construct is a mutant to avoid any confusion.
5. Figure 3A (and B), this experiment needs further clarification. Are these cell extract of streptavidin purification. What is Strep, which cells were used, which is the chimeric protein shown (it seems to be GRB2 but it is not mentioned). Why do the levels of SOS change in the different conditions?
6. Figure 3G, the authors must indicate in the figure legend and labels that this shows RNA levels.
7. Figure 4C, in the image provided it is difficult to see if there is really a decrease of the interaction as claimed by the authors. Do they have a quantification or better image?
8. Figure 4D: What are the authors showing here? Is this RNA or protein levels. It is not indicated in the figure or text, and it is not possible for this reviewer to get this information from the material and methods. Additionally, the authors see a variation of hnRNPC and conclude that this must be regulating GRB3 levels in this samples. This is an important point in the working model but there is not actually real support in the experimental data presented in the manuscript. The authors should show if there is a correlation a correlation with the levels if expression of GRB3-3 in these tumours. They could also show if the regulation of hnRNPC levels has any effect in the transcript's levels of GRB3-3 in cell lines. Otherwise, the authors must tone down their claims in the discussion.

Reviewer #3 (Remarks to the Author):

Growth factor receptor-bound protein 2 (Grb2) is a trivalent adaptor protein and a key element in signal transduction. It interacts via its flanking NSH3 and CSH3 domains with the proline-rich domain of the RasGEF Sos1 and via its central SH2 domain with phosphorylated tyrosine residues of receptor tyrosine kinases (RTKs). The elucidation of structural organization and mechanistic insights into GRB2 interactions, however, remains challenging due to their inherent flexibility. Ketchen et al. describe in this manuscript the cellular role of Grb3-3, a human isoform of Grb2. Grb3-3 arises through alternate splicing of exon four of the Grb2 gene. This results in a 40-amino

acids deletion of the Grb2 SH2 domain, that obviously corrupts the SH2 domain, abrogates pY binding, and hence recruitment to RTKs. The authors show that the SH3 domains remain capable of binding proline-rich sequences, and suggest that Grb3-3 act as a suppressor of Grb2 function.

This study points to the complex biochemical and biophysical issue of cellular control mechanisms. Below are a few of many points that require improvement:

- A major criticism of this study is that the author did not perform any experiment in cells endogenously expressing Grb3-3. How should we accept the importance of Grb3-3 as a natural counterpart of Grb2? Overexpression or deletion constructs of most genes lead to a scenario described in this manuscript: As a result, 'abundantly' expressed gene, such as Grb3-3, competes with the endogenous counterpart, such as the endogenous Grb2, which exists at much lower concentration. Consider this fact then the title of the manuscript must be inverted to 'Grb2 is a positive regulator of RAS activation, as published 30 years ago. Even this cannot be concluded because a direct competition of Grb3-3 with Grb2 has not been shown; so, the statement 'Grb3-3 act as a suppressor of Grb2 function' has not been shown. In summary, the proposed model in Figure 5 on Grb3-3 expressing cells is quite nice but the results do not reflect this conclusion.

- Different key publications, which could clearly improve this manuscript's quality, clarity and transparency, are not considered: (1) Marasco and coworkers have shown in 2000 (doi: 10.1006/bbrc.2000.3415 and 10.1074/jbc.M005535200) that the up-regulation of apoptosis-associated Grb3-3 in HIV-1-infected T-cells and human CD4(+) lymphocytes. (2) Romero et al. have shown that the apoptotic isoform Grb3-3 associates with heterogeneous nuclear ribonucleoprotein C, and these interactions are modulated by poly(U) RNA (DOI: DOI: 10.1074/jbc.273.13.7776). (3) Kazeminejad et al. (doi: 10.1042/BCJ20210105) recently reported an important advance in our mechanistic understanding of how Grb2 allosterically links RTKs to Sos1. This study provides an excellent basis for discussing the data on GRB3-3 and may explain the role of the SH2 domain in blocking CSH3 interaction with Sos1. (4) Other studies have physically connected Grb3-3 to the Rho GTPase regulator Vav and also to Adenosine desamidase.

- The conclusion that Grb3-3 suppresses Grb2 function is based on the data that no GTP-Ras and low levels of p-Erk was observed in Fig. 1a and 1c. However, these data are not consistent with those shown in Figure 3d and 3e, which show that Sos1 is still in complex with Grb2 at the membrane, from which one would expect both Ras activation and Erk phosphorylation. Moreover, large amount of Sos1 is in the membrane fraction in the presence of overexpressed Grb3-3? This is unexplained! Moreover, the rationale for using Caco-2 cells is not clear, particularly because the p-Erk/Erk data in Caco-2 cells (Fig. 1e) are quite different from the data in HEK-293T cells (Fig. 1a).

- Grb2 overexpression and co-expression with Grb3-3 is required under the same condition in Figures 1a-1f to underpin the conclusions.

- Page 5: The statement "alternate splicing of GRB2 is regulated by heterogeneous nuclear ribonucleoproteins C1/C2 (hnRNPC) which suggests a potential therapeutically relevant route to exert control over Grb2/Grb3-3 expression." is unclear. How could Grb3-3 function be used as a potential therapeutic strategy?

Our response to the Reviewers' comments are shown in blue.

Reviewers' comments:

Reviewer #1 (Remarks to the Author):

This is a nice study that demonstrates that expression of a Grb2 splice variant Grb3-3 results in decreased activation of the Ras-MAPK pathway and that this is due competition with Grb2 for SOS binding resulting in a cytoplasmic complex sequestered away from being able to activate Ras. The potential clinical relevance of this splice variant is demonstrated by seeing reduced Grb3-3 levels in tumour vs normal tissue ie. suggestive of a mechanism for enhancing Ras pathway activity in tumours that are otherwise wild-type for Ras. A screen for mediators of splicing identified hnRNPC and this gene is downregulated in tumour vs associated normal tissue.

The first area of clear novelty in this study is the idea of and evidence for competition between Grb3-3 vs Grb2 for SOS - it provides a mechanistic basis for previous observations that are in essence the same as those here showing downregulation of Ras-MAPK signalling. NB. in the 1994 Fath et al Science paper where the effects on RTK-Ras signalling were first observed they relied on functional assays rather than the precise immuno-reagents that are now available but the conclusions are the same. This earlier paper also identified the effect on SH2-mediated interactions and speculated about effects on SOS regulation although the mechanism was not defined.

The profiling of tumour vs normal is also an advance on this previous work although it should be noted this is circumstantial evidence - it is not definitively demonstrated in the clinical samples that there are differences in Ras-MAPK output that correlate with Grb3-3 expression.

The second half of the paper involves a well conducted screen for splicing mediators responsible for Grb3-3 expression. hnRNPC is the lead candidate that is shown to bind to a relevant Grb2 oligo although I note the absence of formal validation that manipulations of hnRNPC expression/function result in concomitant changes in Grb3-3 expression. hnRNPC has previously been shown to bind to Grb2 and Grb3-3 proteins (Romero et al 1998) and so a comment on this in the discussion would be beneficial ie. could this be a mechanism for positive/negative regulation?

In summary the work is nicely performed (inc. appropriate stats analysis). I am convinced by the central observations, they represent an advance on previous work by providing both mechanistic insight and potential clinical relevance.

We are very grateful to this Reviewer for his very positive and encouraging review. His (Prof. Ian Prior) comments highlight the strengths of the manuscript and we are happy that he has picked out areas of novelty, including the evidence for competition between

Grb2 and Grb3-3 for binding to Sos as well as “the well-conducted screen for splicing mediators responsible for Grb3-3 expression.”

Edits/further work

I'm satisfied with the paper as it stands although if the clinical sample correlations and/or the hnRNPC manipulations could be performed they would enhance confidence in the observations. Discussion of hnRNPC:Grb2/Grb3-3 protein would be desirable.

In response to this Reviewer's comments, we have added a comment to the Discussion section regarding the binding of hnRNPC to Grb2 and Grb3-3, and the presence of a possible mechanism for positive/negative regulation. Whilst we agree that the sample correlations and further manipulations of hnRNPC would be of additional interest, we do not have the requisite samples available to us and manipulations of hnRNPC are beyond the scope of this work. It is, of course, reassuring that our screen for splicing mediators – which is described as 'well conducted', for which we are grateful – identified hnRNPC, given the evidence for interactions between hnRNPC and both Grb2 and Grb3-3 that this Reviewer has highlighted.

There was a mix up with labelling of Figure 3 panels in the text - Figure 3ef are Figure 3de etc.

We are grateful to the reviewer for noting this error, which we have now attended to.

Ian Prior

Reviewer #2 (Remarks to the Author):

The Grb2 splice variant, Grb3-3, is a negative regulator of RAS activation.

Ketchen et al.

In this manuscript the authors study the role of a splicing variant of the adaptor protein GRB2 in the regulation of the activation of RAS proteins downstream of RTK. The authors show that GRB3-3 can bind the RAS GEF SOS independent of RTK activation. This binding prevents the binding of SOS to RAS due to sequestration of SOS in the cytoplasm. They also show results indicating that GRB2 and GRB3-3 compete for the interaction with SOS. The authors present evidence that GRB3-3 is a putative tumour suppressor which RNA expression is regulated by hnRNP-dependent splicing. The data presented helps expanding the knowledge on the role of GRB2 in RAS/MAPK regulation and in particular sheds light about the role of the isoform GRB3-3 in cancer. However, while the working model seems to be supported by some of the data presented here, there are some problems with the way the data is presented, the description of the experimental approaches and some further experiments/analysis would

increase the support for the author's conclusions. Thus, while the work presented here might be of interest for the readership of Communications biology but improvement are necessary before it can be recommended for publication.

We are very grateful to this Reviewer for their careful reading of this manuscript, and particularly encouraged by their comments regarding the importance of this work in "expanding the knowledge on the role of Grb2 is RAS/MAPK regulation and in particular sheds light about the role of the isoform Grb3-3 in cancer."

Major points:

1. A big problem with the current manuscript is the lack of sufficient description of the material and methods, and lack of information in the figure legends. This complicates the interpretation of some of the results (see below comment about figure 4D). In general, there are no references for the protocols used, there is no description of critical reagents such as lysis and washing buffers, the plasmids used and some steps of the protocols necessary to allow the reproducibility of the data by the community. The description of the MS analysis set up is especially insufficient, and there is not indication if the results have been deposited in any public repository. This has to be corrected.

The Reviewer has highlighted deficiencies in our detail of materials and methods as well as information in the figure legends. We have addressed all of the specific items requested as follows:

- The Materials & Methods section has been extensively remodelled to substantially increase the level of detail relating to the experimental procedures undertaken for this work.
- Each figure legend has been extensively revised in order to provide a more comprehensive overview of the experimental procedures used to generate the data shown within that figure.
- Note that we have also changed supplementary figure 1b in order to standardise the terms used to label the Sos-oligopeptides, so that these are now consistent between figures.

2. Does GRB3-3 regulate EKR levels in the stable cell lines? In figure 1A and E there seems to be a difference on the level of expression of these kinases in the two cell lines. It would be better if the authors showed the levels of EKR activation in figure 1A in the absence of stimuli to know the basal levels in the stable cell lines. Additionally, do the authors have any explanation why HGF does regulate levels of ERK expression in figure 1A?

Currently we do not have any reason to suspect that the presence of Grb3-3 regulates ERK levels in stable cells. Figure 1e shows that in the CaCo-2 cells comparison of bands

blotted for ERK in control and transfected cells reveals that Grb3-3 has no apparent impact on ERK expression. We would not expect that the expression of ERK should be constant in the two cell lines adopted in the two experiment shown in Figure 1a and 1e, i.e. HEK293T and CaCo-2 respectively.

We do not believe that HGF regulates the level of ERK expression, since growth factor stimulation was limited to a short (5 minute) time course. The differences of ERK levels in the bands in Figure 1a are due to sample fluctuation. Importantly, the level of pERK is normalised against these total ERK levels for each experiment (as seen in Figure 1b) to demonstrate the impact of Grb3-3 expression.

3. Figure 1C, the pull down should be quantified against the total levels of RAS not GST-RBD. This is the common way of presenting this kind of experiments. It is also common to present this as % of activation

(i.e. <https://www.sciencedirect.com/science/article/pii/S2451945617303239#fig4>). The inclusion of a blot with RAS expression levels in cell extract would also increase confidence that no changes shown are due to a possible regulation of RAS level by expression of GRB3-3

This experiment was designed to show the activation of RAS through the ability of the active, GTP-bound form to bind to the RBD of RAF-1. These experiments were done in the same cell line using the same conditions, i.e. transfection, serum starvation and stimulation. Any impact of differential RAS expression would be expected to be manifested in all of the cells used. Furthermore, the level of transfected GFP-tagged RAS are likely to be high and hence not limiting with respect to availability for GTP binding. As a result the control we adopted was the total GST-RBD because this provides the indirect signal for RAS activation. In the excellent paper from the Shokat lab highlighted by the Reviewer, the quantification against total RAS is important because the blots are for a mutant form of H-RAS and are designed to highlight the binding of a small fragment molecule. Thus, there could be inconsistencies in expression between the experiments which require the RAS expression control.

4. Figure 1G and H, the labels should indicate that the GRB2 construct is a mutant to avoid any confusion.

We are grateful to the reviewer for highlighting this and have amended the Grb2 label in figures 1g and 1h, accordingly.

5. Figure 3A (and B), this experiment needs further clarification. Are these cell extract of streptavidin purification. What is Strep, which cells were used, which is the chimeric protein shown (it seems to be GRB2 but it is not mentioned). Why do the levels of SOS change in the different conditions?

We are grateful to the reviewer for highlighting this and have both (1) included further detail regarding the use of the streptavidin-tag (*Strep*-tag) system for one-step purification in the methods, and (2) included further experimental detail in the relevant figure legend. As is now highlighted within the figure legend, Sos was immunoprecipitated with RFP-tagged Grb2, and reduces in the presence of exogenous Grb3-3; i.e. reflecting competition between Grb2 and Grb3-3 for Sos binding.

6. Figure 3G, the authors must indicate in the figure legend and labels that this shows RNA levels.

We have amended both the figure and the figure legend in response to the reviewer's helpful suggestion.

7. Figure 4C, in the image provided it is difficult to see if there is really a decrease of the interaction as claimed by the authors. Do they have a quantification or better image?

We have added in densitometry to this figure (Fig 4d), as requested.

8. Figure 4D: What are the authors showing here? Is this RNA or protein levels. It is not indicated in the figure or text, and it is not possible for this reviewer to get this information from the material and methods.

We are grateful to the Reviewer for highlighting that this had been inadequately described. We have addressed this and modified the Materials and Methods section to clarify.

Additionally, the authors see a variation of hnRNPC and conclude that this must be regulating GRB3 levels in this samples. This is an important point in the working model but there is not actually real support in the experimental data presented in the manuscript. The authors should show if there is a correlation a correlation with the levels if expression of GRB3-3 in these tumours. They could also show if the regulation of hnRNPC levels has any effect in the transcript's levels of GRB3-3 in cell lines. Otherwise, the authors must tone down theirs claims in the discussion.

We are grateful to the reviewer for this critique of our data, and we think this comment is broadly fair. As such, we have refashioned our Discussion to make explicit mention of the lack of a direct correlation between hnRNPC expression and corresponding Grb3-3 expression in the data available to us. We have in addition expanded on the Discussion by including more information relating to previous studies suggesting an interaction between hnRNPC and both Grb2 and Grb3-3, which as reviewer 1 highlights supports our data and allows us to speculate on a potential mechanistic basis for the feedback mechanism we have proposed.

Reviewer #3 (Remarks to the Author):

Growth factor receptor-bound protein 2 (Grb2) is a trivalent adaptor protein and a key element in signal transduction. It interacts via its flanking NSH3 and CSH3 domains with the proline-rich domain of the RasGEF Sos1 and via its central SH2 domain with phosphorylated tyrosine residues of receptor tyrosine kinases (RTKs). The elucidation of structural organization and mechanistic insights into GRB2 interactions, however, remains challenging due to their inherent flexibility. Ketchen et al. describe in this manuscript the cellular role of Grb3-3, a human isoform of Grb2. Grb3-3 arises through alternate splicing of exon four of the Grb2 gene. This results in a 40-amino acids deletion of the Grb2 SH2 domain, that obviously corrupts the SH2 domain, abrogates pY binding, and hence recruitment to RTKs. The authors show that the SH3 domains remain capable of binding proline-rich sequences, and suggest that Grb3-3 act as a suppressor of Grb2 function.

This study points to the complex biochemical and biophysical issue of cellular control mechanisms. Below are a few of many points that require improvement:

We would like to thank this Reviewer for their considered comments on this manuscript. Clearly the Reviewer has significant experience in this area and familiarity of the Grb2-Sos-RAS signal transduction mechanism. This knowledge is reflected in many of the comments, however there does seem to be an inadvertent misrepresentation of our data in some of the comments.

- A major criticism of this study is that the author did not perform any experiment in cells endogenously expressing Grb3-3. How should we accept the importance of Grb3-3 as a natural counterpart of Grb2 Overexpression or deletion constructs of most genes lead to a scenario described in this manuscript: As a result, 'abundantly' expressed gene, such as Grb3-3, competes with the endogenous counterpart, such as the endogenous Grb2, which exists at much lower concentration. Consider this fact then the title of the manuscript must be inverted to 'Grb2 is a positive regulator of RAS activation, as published 30 years ago. Even this cannot be concluded because a direct competition of Grb3-3 with Grb2 has not been shown; so, the statement 'Grb3-3 act as a suppressor of Grb2 function' has not been shown. In summary, the proposed model in Figure 5 on Grb3-3 expressing cells is quite nice but the results do not reflect this conclusion.

It is not entirely clear what the Reviewer is asking for in this paragraph. The reference to using cells endogenously expression Grb3-3 is somewhat puzzling. The cell-based part of this study is focused on a mechanistic understanding of the role of Grb3-3. To ascertain this, the approach requires a model cell line in which we can demonstrate the impact of expression of the splice variant. To demonstrate any inhibitory effect of Grb3-3 compared with Grb2 on RAS signalling we need to perturb the cellular concentrations of

Grb3-3 in a detectable manner. The presence of endogenous Grb3-3 would complicate these studies.

From a technical standpoint the endogenous expression levels of Grb2 and Grb3-3 are not easily ascertained and compared because of the absence of an antibody that recognises Grb3-3. We attempted to make such an antibody directed against an epitope crossing the boundary of the 40 residue deletion without success.

Furthermore, the Reviewer is aware that we have demonstrated the endogenous expression of Grb3-3 in colon tissue, and we have even gone so far as to show that it is expressed at reasonable relative percentages with respect to Grb2 (as measured by the total gene output) in tissue.

The question "How should we accept the importance of Grb3-3 as a natural counterpart of Grb2" seems strange based on the existing evidence for Grb3-3 being a suppressor of Grb2 function (as referenced by Reviewer 1). It is also not clear what point the Reviewer is making in saying "Overexpression of deletion constructs of most genes lead to a scenario described in this manuscript". We show specifically that the Grb3-3 expression affects RAS activation and downstream response. It is not clear whether the Reviewer is suggesting that deletion of other regions of Grb2 would have the same effect. This clearly would not be the case since the presence of the intact SH3 domain(s) in GRB3-3 are required for the sequestration of Sos from recruitment to RAS.

We would argue that our data do indeed show that Grb3-3 is a "negative regulator" of RAS activation and that the mechanism proposed in the light of our data, and other mechanistic studies on Grb2 function (some referenced by this Reviewer) is robust.

- Different key publications, which could clearly improve this manuscript's quality, clarity and transparency, are not considered: (1) Marasco and coworkers have shown in 2000 (doi: 10.1006/bbrc.2000.3415 and 10.1074/jbc.M005535200) that the up-regulation of apoptosis-associated Grb3-3 in HIV-1-infected T-cells and human CD4(+) lymphocytes. (2) Romero et al. have shown that the apoptotic isoform Grb3-3 associates with heterogeneous nuclear ribonucleoprotein C, and these interactions are modulated by poly(U) RNA (DOI: DOI: 10.1074/jbc.273.13.7776). (3) Kazeminejad et al. (doi: 10.1042/BCJ20210105) recently reported an important advance in our mechanistic understanding of how Grb2 allosterically links RTKs to Sos1. This study provides an excellent basis for discussing the data on GRB3-3 and may explain the role of the SH2 domain in blocking CSH3 interaction with Sos1. (4) Other studies have physically connected Grb3-3 to the Rho GTPase regulator Vav and also to Adenosine desamidase.

We should like to thank the Reviewer for highlighting additional literature relevant to the Grb3-3 and Grb2 field. Many of these excellent studies provide a general back-drop to

the field and indeed the work by *Romero et al* was already recognised in our discussion, though we have emphasised it further in line with the importance stressed by the reviewers. We have in addition expanded our discussion to include a wider 'non-cancer' paragraph reflecting the wider implications of our findings, through which we have incorporated the reviewer's valuable suggestions.

- The conclusion that Grb3-3 suppresses Grb2 function is based on the data that no GTP-Ras and low levels of p-Erk was observed in Fig. 1a and 1c. However, these data are not consistent with those shown in Figure 3d and 3e, which show that Sos1 is still in complex with Grb2 at the membrane, from which one would expect both Ras activation and Erk phosphorylation. Moreover, large amount of Sos1 is in the membrane fraction in the presence of overexpressed Grb3-3? This is unexplained!

The Reviewer makes an important point here, and one that deserves clarification. As the Reviewer suggests our data in Figure 1 show that the presence of Grb3-3 down-regulates the phosphorylation of ERK and activation of RAS. Our data strongly point to a mechanism in which the binding of Grb3-3 to Sos via its intact SH3 domain(s) and the inability of Grb3-3 to interact with RTKs means that the availability of Sos for binding to membrane-localised RAS is reduced. This is reflected in the fractionation experiments in Figure 3d and e. We would not anticipate that Grb3-3 would be capable of sequestering every Sos molecule in the cytoplasm. Therefore, we would expect to see a reduction in Sos at the membrane in the presence of Grb3-3 but not complete abrogation of membrane localisation. Conversely, we do also see increased concentration of Sos in the cytoplasm in the presence of GRB3-3 as we would predict from the above mechanism.

Moreover, the rationale for using Caco-2 cells is not clear, particularly because the p-Erk/Erk data in Caco-2 cells (Fig. 1e) are quite different from the data in HEK-293T cells (Fig. 1a).

The use of Caco-2 cells provide a different cell line exemplar of the impact of Grb3-3 as well as them having some relevance to our later studies on CRC patients.

- Grb2 overexpression and co-expression with Grb3-3 is required under the same condition in Figures 1a-1f to underpin the conclusions.

In all experiments, the cell lines adopted have endogenous expression of Grb2. In the absence of this the initiation of the RAS signalling pathway and downstream phosphorylation in response to RTK stimulation would be compromised. Therefore, against this existing backdrop, through overexpression of Grb3-3 we can demonstrate the inhibitory effect on this endogenous expression. As a result we would suggest that additional expression of exogenous Grb2 would provide little additional clarity to our mechanistic study. Indeed, it might lead to additional effects and initiation of off target signalling. Indeed, one might envisage an experiment whereby we attempt to demonstrate competition between Grb2 and Grb3-3 in cell using 'titration' of Grb2-

containing plasmid into cells already transfected with Grb3-3. Although, based on our biophysical studies of the respective K_d 's of the proteins this should be possible, the downstream readout of RAS activation could be influenced by off-target binding of Grb2 to other ligands in the cell. In addition, we hope that Figure 3g goes some way to reassuring the Reviewer in terms of demonstrating significant levels of Grb3-3 gene output in both normal and malignant cells.

- Page 5: The statement "alternate splicing of GRB2 is regulated by heterogenous nuclear ribonucleoproteins C1/C2 (hnRNPC) which suggests a potential therapeutically relevant route to exert control over Grb2/Grb3-3 expression." is unclear. How could Grb3-3 function be used as a potential therapeutic strategy?

We apologise to the Reviewer for not clarifying this point. The Reviewer is likely to be aware of the growing interest in the adoption RNA-based therapies directed at modification of transcriptional outputs either through influence of transcription itself or modification of protein expression profiles through miRNAs/mRNAs.

Reviewers' comments:

Reviewer #1 (Remarks to the Author):

I'm generally satisfied. The data are consistent with the model, it's just now a case of whether this really happens in real life - as alluded to by reviewer 3. The mRNA analysis of Grb2 vs Grb3-3 gave me confidence that there is expression of Grb3-3 but it is still unclear if the amounts of Grb3-3 protein expressed will be sufficient to have meaningful effects on Ras signalling. The authors were unable to firm up their conclusions with additional data. The authors have been fair in their discussion in caveating some of the shortcomings - could also caveat the lack of formal measurement of endogenous Grb3-3 protein and what experiments would ideally be conducted to quantify Grb2:Grb3-3 to see if competition is likely to be relevant in the endogenous context.

Reviewer #2 (Remarks to the Author):

The authors have addressed most of the point I raised and the manuscript is much improved. I consider that the models and experimental approach are in general correct for testing the hypotheses of the different sections. The model is now more realistically discussed and the experimental description is excellent. Thus, I do recommend the publication of the current manuscript

Reviewer #3 (Remarks to the Author):

The revised manuscript has improved significantly. I am fine with the authors' responses. However, two aspects remain unclear about which the authors may wish to rethink:

(a) if "Grb3-3 act as a suppressor of Grb2 function" then one would expect complete abrogation of SOS recruitment and activation cloth Grb2. And this is not the case, even not under over expression conditions, where Grb3-3 concentration is higher above the Grb2 concentration.

(b) It is understandable if studies that address RAS signaling in any way are associated with therapeutic interventions (regardless of how realistic the proposals are). However, the function of Grb3-3 as a potential "therapeutic strategy" appears to me to be quite difficult to realize due to the sequence identity of Grb2 and Grb3-3 at both the protein and RNA levels. Adding a short, clarifying text on this topic would be useful.

Our responses to the reviewers' comments are shown in green:

Editor's comments

We therefore ask that you provide the uncropped and unprocessed images for each blot/gel in the Supplementary Information. We also ask that loading controls are provided for each gel when run separately.

Unprocessed images for each blot/gel are now provided in Extended Data Figure 4 and Extended Data Figure 5. We have also improved the resolution of some of the blots included within the various figure panels that make up this manuscript.

For all instances of splicing, we also require that you indicate the splicing by adding a line in the Figure, and explaining in each legend what the line represents and how the experiment was done (i.e. if the samples were run on the same gel or different gels, under the same conditions etc).

We are not sure exactly what this refers to. Perhaps there is a misunderstanding. We did not include any novel gene splicing in our experiments. The only spliced variant is the Grb3-3.

Reviewer #1 (Remarks to the Author):

I'm generally satisfied. The data are consistent with the model, it's just now a case of whether this really happens in real life - as alluded to by reviewer 3. The mRNA analysis of Grb2 vs Grb3-3 gave me confidence that there is expression of Grb3-3 but it is still unclear if the amounts of Grb3-3 protein expressed will be sufficient to have meaningful effects on Ras signalling. The authors were unable to firm up their conclusions with additional data. The authors have been fair in their discussion in caveating some of the shortcomings - could also caveat the lack of formal measurement of endogenous Grb3-3 protein and what experiments would ideally be conducted to quantify Grb2:Grb3-3 to see if competition is likely to be relevant in the endogenous context.

We are grateful to Reviewer #1 for their appraisal of our manuscript and we are equally pleased that they both see the value in the Grb2 vs Grb3-3 mRNA analysis and agree with the fairness of our discussion section.

We note the comment regarding whether there would be sufficient amounts of Grb3-3 protein expressed to have meaningful effects on Ras signalling. In response, we would highlight that in their 1994 *Science* paper, in which they provided the first evidence for *Grb3-3*, *Fath et al* (referenced within our manuscript) identified that *Grb3-3* is widely expressed. Equally, and as now referenced in our discussion, multiple other studies have shown sufficiently large amounts of Grb3-3 to compete with Grb2 (see refs 13, 22, 23). It is also, in our opinion, not so much in doubt that Grb3-3 can impact on the Ras pathway. *Fath et al* demonstrated this by showing that EGF-dependent transactivation of Ras-responsive element (RRE) in Chinese hamster lung fibroblasts was increased by 40% with exogenous *Grb2* transfection, and decreased by 50% with *Grb3-3* transfection. Interestingly, when expressed at a 1:1 ratio with Grb2, *Grb3-3* abrogated EGF-induced RRE activation, which we have provided a mechanistic basis for in the manuscript provided here. Reviewer #1

helpfully highlights the need for this to be more explicitly addressed in our manuscript, which we have now done within the discussion section.

Reviewer #2 (Remarks to the Author):

The authors have addressed most of the point I raised and the manuscript is much improved. I consider that the models and experimental approach are in general correct for testing the hypotheses of the different sections. The model is now more realistically discussed and the experimental description is excellent. Thus, I do recommend the publication of the current manuscript

We are grateful for Reviewer #2 for their positive appraisal of the changes we have made to the manuscript in response to feedback.

Reviewer #3 (Remarks to the Author):

The revised manuscript has improved significantly. I am fine with the authors' responses.

However, two aspects remain unclear about which the authors may wish to rethink:

(a) if "Grb3-3 act as a suppressor of Grb2 function" then one would expect complete abrogation of SOS recruitment and activation cloth Grb2. And this is not the case, even not under over expression conditions, where Grb3-3 concentration is higher above the Grb2 concentration.

We agree that this could have been worded better, and have adjusted the phrasing in the final paragraph of page 4 accordingly. We are grateful to the reviewer for highlighting the potential for this to be misinterpreted.

(b) It is understandable if studies that address RAS signaling in any way are associated with therapeutic interventions (regardless of how realistic the proposals are). However, the function of Grb3-3 as a potential "therapeutic strategy" appears to me to be quite difficult to realize due to the sequence identity of Grb2 and Grb3-3 at both the protein and RNA levels. Adding a short, clarifying text on this topic would be useful.

We agree that this needs to be carefully considered and are grateful for the reviewer highlighting this. Given the potential difficulties in the therapeutic strategy we suggested, and so as not to detract from the main message of the manuscript, we have adjusted the final paragraph to instead focus on the role of the Grb2:Grb3-3 ratio in disease states such as cancer, rather than detail on specific potential therapeutic intervention.

REVIEWERS' COMMENTS:

Reviewer #1 (Remarks to the Author):

I have no further comments. The article has been suitably revised in response to comments.

Reviewer #3 (Remarks to the Author):

The authors have addressed all of the points. The manuscript is sound, now.